# Manipulations of multi-frequency waves and signals via multi-partition asynchronous space-time-coding digital metasurface

Si Ran Wang [1,2,3,5], Jun Yan Dai [1,2,3,5], Qun Yan Zhou[1,2,3], Jun Chen Ke[1,2,3], Qiang Cheng [1,2,3] ✉ & Tie Jun Cui [1,2,3,4] ✉

Manipulations of multiple carrier frequencies are especially important in a variety of fields like radar detection and wireless communications. In conventional radio-frequency architecture, the multi-frequency control is implemented by microwave circuits, which are hard to integrate with antenna apertures, thus bringing the problems of expensive system and high power consumption. Previous studies demonstrate the possibility to jointly control the multiple harmonics using space-time-coding digital metasurface, but suffer from the drawback of inherent harmonic entanglement. To overcome the difficulties, we propose a multi-partition asynchronous space-time-coding digital metasurface (ASTCM) to generate and manipulate multiple frequencies with more flexibility. We further establish an ASTCM-based transmitter to realize wireless communications with frequency-division multiplexing, where the metasurface is responsible for carrier-wave generations and signal modulations. The direct multi-frequency controls with ASTCM provides a new avenue to simplify the traditional wireless systems with reduced costs and low power consumption.

The capability of independent manipulations of multi-frequency signals is of great importance to modern wireless systems, due to the widespread use of multi-carrier technologies in a wide range of application scenarios. For instance, the well-known orthogonal frequency division multiplexing (OFDM) offers an efficient sub-carrier broadband modulation technique in mobile communication systems[1–5], giving rise to enhanced data-throughput capacity by increasing the number of non-interfering channels. In radar detections, the frequency diverse array (FDA) that introduces tiny frequency increments across the internal elements has shown remarkable capability to create angle-range-dependent transmitting beampatterns for mitigating multipath interferences, which is especially favored to suppress the range ambiguous clutter compared to the conventional active phased array[6–10].

So far, the efficient controls of multi-frequency signals have been widely demonstrated at the circuit level in traditional electronic systems, where a plurality of carrier frequencies is produced from the local oscillators (LOs), with the amplitude and phase of each frequency component dynamically adjusted by using radio-frequency (RF) amplifiers and phase shifters[11]. However, RF devices are usually characterized by high loss, high integration complexity, and large cost at high frequencies, and thus severely hinder their applications in practice.

Recently, significant advances on wave-matter-information interactions with space-time-coding digital metasurface (STCM) have been triggered by its capabilities to manipulate the electromagnetic (EM) waves in both spatial and temporal domains[12–31]. The customization of intrinsic reflection and/or transmission

[1]State Key Laboratory of Millimeter Waves, Southeast University, Nanjing 210096, China. [2]Institute of Electromagnetic Space, Southeast University, Nanjing 210096, China. [3]Frontiers Science Center for Mobile Information Communication and Security, Southeast University, Nanjing 210096, China. [4]Pazhou Laboratory, Huangpu 510555 Guangzhou, China. [5]These authors contributed equally: Si Ran Wang, Jun Yan Dai. ✉e-mail: qiangcheng@seu.edu.cn; tjcui@seu.edu.cn

characteristics in each period for STCM offers the possibility of generating multiple harmonics at the same time with high nonlinear conversion efficiencies[32–34]. Moreover, the strategy for joint amplitude and phase modulations of the nonlinear harmonics has been presented to realize precise wave manipulations at different harmonic frequencies, thereby giving rise to multiple wireless channels for information transmissions[23,35–40]. In general, the continuous reflection phase is discretized into finite phase states with the phase step of $2\pi/2^n$, where $n$ is the quantization bit number. However, owing to the inherent harmonic entanglement[41], the amount of simultaneously regulated harmonics is largely determined by the quantification bit number of the reflection phase. Since the multi-frequency manipulations require to decompose the temporal coding sequence into a series of intertwined sub-sequences, the regulation speed is greatly hampered by the coding complexity.

Aiming at this limitation, an asynchronous space-time-coding digital metasurface (ASTCM)[42] was further developed, where the whole aperture is divided into several sub-arrays with different modulation frequencies under the same excitation. In this case, the independently generated harmonic number is no longer constrained to the phase quantification of the meta-atom, hereby enhancing the nonlinear manipulation capability of the metasurface. The interferences of various harmonics are especially beneficial for applications like automatic scanning antennas and dynamic radar cross sections, but the joint manipulations of the multiple harmonics and the corresponding beam-forming method remain a challenging task to achieve the full potentials of ASTCM.

On this basis, here we propose a coding strategy for controlling the multi-frequency signals with the aid of multi-partition ASTCM, which allows independent reconfigurations of the EM-wave amplitude, phase, and scattering pattern for all frequency components in a predefined manner. A prototype of simplified-architecture frequency-division multiplexing transmitter is designed and realized to implement wireless signal transmissions. Eight carrier frequencies are employed in this system to demonstrate the capability of the proposed wave manipulation method. We believe that more sub-carriers could be achieved to expand the communication capacities further of the wireless systems, bringing more potentials of the next-generation wireless communications.

## Results

### Background theory

As sketched in Fig. 1, a multi-partition ASTCM is employed to generate and manipulate the propagation features of multiple nonlinear harmonics simultaneously in real-time. In order to better elucidate the underlying physics, we start from a simple scenario where the metasurface is only composed of two partitions that are responsible for the manipulation of dual frequencies respectively. From the inset panels of Fig. 1, we can see that the sizes and shapes of partitions can be reconfigured in real-time on the metasurface to meet the demands of engineering the nonlinear harmonics and shaping the nonlinear wavefronts.

Under the illumination of a monochromatic wave along normal direction, the embedded diodes of the meta-atoms in different partitions can be controlled by periodic voltage signals with distinct modulation frequencies[42], and therefore the reflection spectra are featured by two nonlinear harmonics. For simplicity, we assume that there are two partitions $1^{\#}$ and $2^{\#}$ on the metasurface. The incident frequency is $f_c$, and the desired reflecting frequencies are $f_1$ and $f_2$. In addition, the desired harmonic fields from the partitions $1^{\#}$ and $2^{\#}$ can be written as:

$$\begin{cases} E_1^m = A_m e^{j\varphi_m} e^{j2\pi f_1 t} \\ E_2^n = A_n e^{j\varphi_n} e^{j2\pi f_2 t} \end{cases}, \tag{1}$$

where the subscript and superscript of $E$ represent the harmonic order and partition order, respectively, $(A_m, \varphi_m)$ and $(A_n, \varphi_n)$ are the corresponding harmonic amplitudes and phases, and $\Delta f_1$ and $\Delta f_2$ are the modulation frequencies of the two partitions. Different combinations of $(m, n, \Delta f_1, \Delta f_2)$ can be selected to achieve:

$$f_1 = f_c + m\Delta f_1 \tag{2}$$

$$f_2 = f_c + n\Delta f_2. \tag{3}$$

Aiming to synthesize the harmonic amplitudes and phases at $f_1$ and $f_2$, it is essential to customize the periodic temporal reflectivity function of the metasurface, where two reflection phase states 0 and $\pi$ are switched alternatively. The period of the reflectivity function is

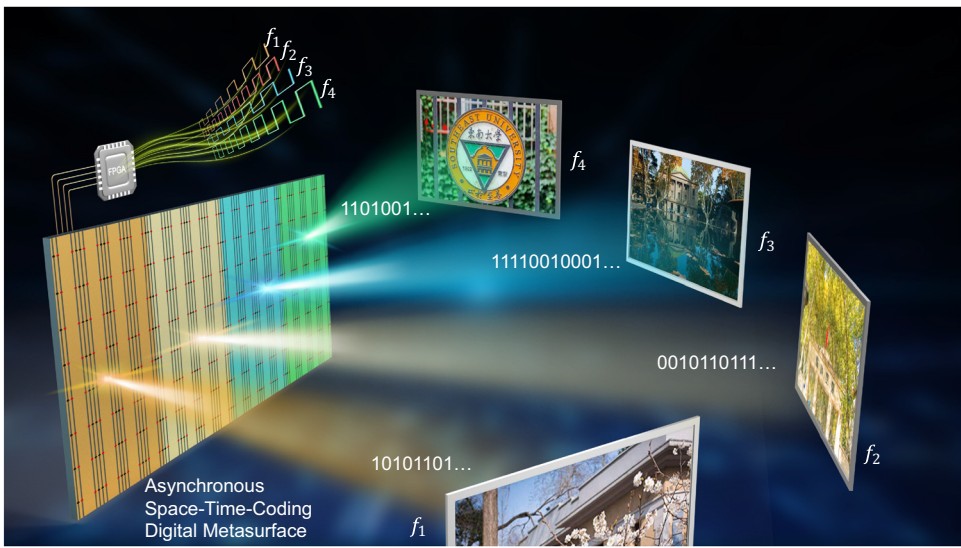

**Fig. 1 | Schematic of wave manipulations at multiple frequencies based on the asynchronous space-time-coding digital metasurface.** The partitions of the metasurface with different colors are responsible for the wave manipulations at different carrier frequencies ($f_1, f_2, f_3, f_4$).

$T = 1/\Delta f$, and the duration is $\tau$ for the phase 0 in each period. Therefore, the duty ratio becomes $M = \tau \cdot \Delta f$. According to the theory of Fourier Series[33], the complex amplitude $A_k e^{j\varphi_k}$ of the $k^{th}$-order harmonic can be expressed as:

$$A_k e^{j\varphi_k} = \frac{1}{T} \int_0^T \Gamma(t) e^{-j2\pi k \Delta f t} dt, \qquad (4)$$

in which $\Gamma(t)$ represents the time-varying reflectivity of the metasurface, and $A_k$ and $\varphi_k$ can be given by[37]:

$$\begin{cases} A_k = 2 \cdot M \cdot \left| \frac{\sin k\pi M}{k\pi M} \right|, \\ \varphi_k = -\frac{\pi}{2} \left[ 1 - (-1)^{\lfloor |k| \cdot M \rfloor} \right]. \end{cases} \quad k = \pm 1, \pm 2, \ldots, \qquad (5)$$

Here, $\lfloor \rfloor$ indicates the operation of rounding down. From Eq. (5), it is evident that in this case both amplitude and phase of the $k^{th}$-order harmonic are closely related to the duty ratio $M$. Furthermore, from Eqs. (1)–(4) we have:

$$\begin{cases} A_m e^{j\varphi_m} = \frac{1}{T_1} \int_0^{T_1} \Gamma_1(t) e^{-j2\pi m \Delta f_1 t} dt \\ A_n e^{j\varphi_n} = \frac{1}{T_2} \int_0^{T_2} \Gamma_2(t) e^{-j2\pi n \Delta f_2 t} dt \end{cases}, \qquad (6)$$

in which $T_1 = 1/\Delta f_1$ and $T_2 = 1/\Delta f_2$ represent the modulation periods of the partitions 1# and 2#, respectively; and $\Gamma_1(t)$ and $\Gamma_2(t)$ refer to the temporal reflectivity of the partitions 1# and 2#, respectively. To further improve the ability of harmonic phase control, an extra time delay can be introduced into the original temporal reflectivity. Specifically, when the time delays $\Delta t_1$ and $\Delta t_2$ are respectively applied to $\Gamma_1(t)$ and $\Gamma_2(t)$, the amplitudes and phases of the two carriers can be rewritten as[34]:

$$\begin{cases} A_m e^{j(\varphi_m - 2\pi m \Delta f_1 \Delta t_1)} = \frac{1}{T_1} \int_0^{T_1} \Gamma_1(t - \Delta t_1) e^{-j2\pi m \Delta f_1 t} dt \\ A_n e^{j(\varphi_n - 2\pi n \Delta f_2 \Delta t_2)} = \frac{1}{T_2} \int_0^{T_2} \Gamma_2(t - \Delta t_2) e^{-j2\pi n \Delta f_2 t} dt \end{cases}. \qquad (7)$$

From Eq. (7), it can be seen that the time delays only affect the harmonic phases rather than the amplitudes, which enables the separation of the two responses effectively. Therefore, we are able to synthesize the desired two harmonics at $f_1$ and $f_2$ by selecting the combinations of $\Gamma_1(t)$ and $\Gamma_2(t)$ with proper combinations of the duty ratios $M_1$, $M_2$, and the time delays $\Delta t_1$, $\Delta t_2$. More discussions on selecting $M_1$, $M_2$, $\Delta t_1$, and $\Delta t_2$ can be found in Supplementary Information Note 1.

By letting $\Gamma_1(t) = \Gamma_2(t) = \Gamma_0(t)$, and $T_1 = T_2 = T$ in Eq. (6), and the harmonic amplitudes and phases can be expressed as follows:

$$\begin{cases} A_m e^{j\varphi_m} = \frac{1}{T} \int_0^T \Gamma_0(t) e^{-j2\pi m \Delta f t} dt \\ A_n e^{j\varphi_n} = \frac{1}{T} \int_0^T \Gamma_0(t) e^{-j2\pi n \Delta f t} dt \end{cases}. \qquad (8)$$

In Eq. (8), we give a traditional way for dual-frequency manipulations using the whole metasurface. Once the reflectivity function $\Gamma_0(t)$ is chosen for the $m^{th}$-order harmonic manipulation, the amplitude and phase of the $n^{th}$ order harmonic are also determined as well. In order to independently control the two harmonics with arbitrary $A_m$, $A_n$, $\varphi_m$ and $\varphi_n$, it poses great challenges to find proper reflectivity function $\Gamma_0(t)$. In fact, it is hard to find or realize such time-varying reflectivity functions according to the desired amplitude and phase combinations. To circumvent this problem, the intertwined coding sub-sequences were employed to generate and manipulate multiple harmonics[41], but it suffers from three additional major problems: complicated coding algorithms, slow reconfiguration rate, and limited harmonic number. In general, the independently controlled harmonic number is equal to the quantized bit width of the reflection phase of the metasurface. This

implies that a large number of PIN diodes are demanded in the meta-atoms to achieve more phase states to promote the number of harmonics, resulting in significant increase of system cost and power consumption. Such a strategy is hard to extend for multiple-frequency manipulations since we need to find a very complex reflection phase function $\Gamma_0$. However, in Eq. (7), we can use different partitions with independent modulation waveforms to circumvent this problem. As a result, the number of independent frequency channels is greatly improved, which is equal to the partition number within the metasurface.

Figure 2 shows the dependence of the complex amplitudes at two frequencies $f_1$ and $f_2$ on the time-varying reflection coefficients $\Gamma_1(t)$ and $\Gamma_2(t)$ of Partitions 1# and 2#. It is clear that by independently designing the square reflection function as investigated in ref. 37, the amplitudes and phases of the two frequency components can be synthesized in an arbitrary manner without mutual coupling. As an example, from Fig. 2a–c, the normalized complex amplitudes at $f_1$ and $f_2$ are ($e^{i\pi/4}$, $e^{-i\pi/4}$), ($\frac{1}{2}e^{i\pi/4}$, $e^{i\pi/2}$) and ($\frac{1}{2}e^{-i\pi/4}$, $\frac{1}{2}e^{i\pi/2}$), respectively, and the corresponding reflection coefficients are demonstrated by black and red lines in Fig. 2d–f. In the calculation, we assume that the incident frequency is 4.25 GHz. The modulation frequencies ($\Delta f_1$, $\Delta f_2$) of the partitions 1# and 2# are 100 kHz and 200 kHz, respectively. In the meanwhile, the orders of harmonics in partitions 1# and 2# are $m = -1$ and $n = +1$ in Eq. (7), so the corresponding reflection frequencies from the two regions are $f_1 = 4.2499$ GHz and $f_2 = 4.2502$ GHz. It is worth noting that by introducing the space-division multiplexing, we only need to change the pulse delay and the duty ratio of the square reflection waveform for dual frequency controls, as illustrated in Fig. 2d–f, without developing the complicated coding schemes employed on the whole metasurface.

The fine control of amplitude and phase greatly facilitates the beam shaping applications at dual frequencies. Since the two partitions are excited by the monochromatic wave under the normal incidence, the far-field scattering patterns of the Partitions $1^{\#}$ and $2^{\#}$ for the $m^{th}$ and $n^{th}$ order harmonics can be written as:

$$\begin{cases} F_1^m(\theta, \varphi) = \sum_{p, q \in 1\#} E_1^m(p, q) e^{j2\pi(f_c + m\Delta f_1)[d_x(p,q)\sin\theta\cos\varphi + d_y(p,q)\sin\theta\sin\varphi]} \\ F_2^n(\theta, \varphi) = \sum_{p, q \in 2\#} E_2^n(p, q) e^{j2\pi(f_c + n\Delta f_2)[d_x(p,q)\sin\theta\cos\varphi + d_y(p,q)\sin\theta\sin\varphi]} \end{cases} \qquad (9)$$

where $\theta$ and $\varphi$ represent the elevation and azimuth angles, $p$ and $q$ denote the element numbers along the $x$ and $y$ directions, $E_1^m(p,q)$ is the far-field scattering pattern of the meta-atom $(p, q)$ in partition 1# at the $m^{th}$ harmonic, $E_2^n(p,q)$ is the far-field scattering pattern of the meta-atom $(p, q)$ in partition 2# at the $n^{th}$ harmonic, and $d_x$ and $d_y$ are element periods along the $x$ and $y$ directions. Here $E_1^m(p,q)$ and $E_2^n(p,q)$ can be adjusted by utilizing different time-varying reflection functions of ASTCM from Fig. 2.

From the antenna theory, different space partitioning methods and the corresponding phase distributions play an important role in the scattering patterns at the two frequencies. To illustrate the mechanism, the dark green and orange regions in Fig. 3 are modulated by the frequencies of 100 kHz and 200 kHz, respectively, and the metasurface is illuminated by a plane wave at 4.25 GHz. As a result, the echo frequencies from the dark green and orange regions are 4.2501 GHz and 4.2502 GHz, respectively. In order to control the scattering beam angle, phase gradient should be introduced among the dark green and orange columns according to the generalized Snell's law. In Fig. 3a, b, the phase distributions from Columns 1 to 16 are 0, $\pi/2$, $\pi$, $3\pi/2$, 0, $\pi/2$, $\pi$, $3\pi/2$, $3\pi/2$, $\pi$, $\pi/2$, 0, $3\pi/2$, $\pi$, $\pi/2$, and 0. According to the formula of scattering far-field pattern, the main lobes of the scattering patterns at $f_1$ and $f_2$ should point to −44°

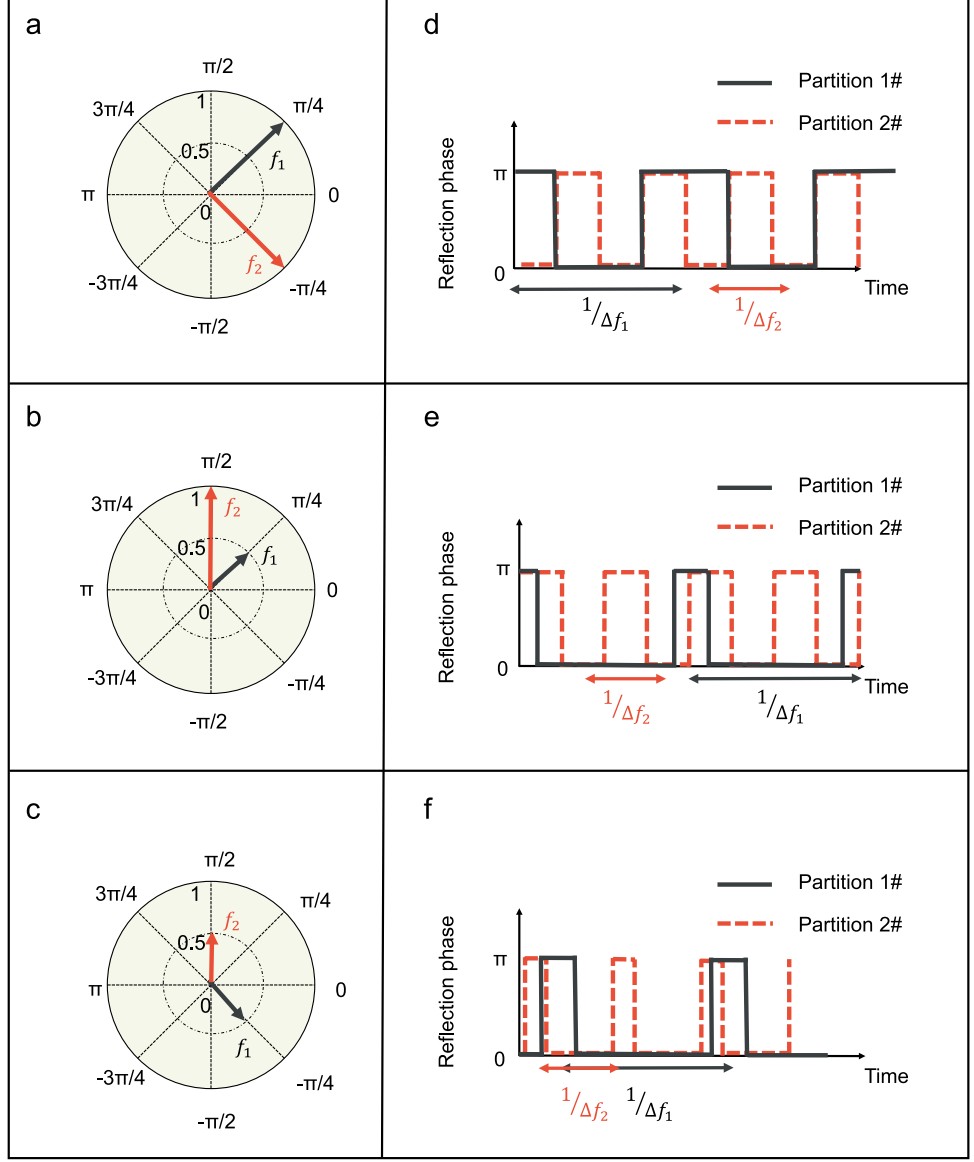

**Fig. 2 | Phasor diagrams for the frequency components $f_1$ (black arrow line) and $f_2$ (red arrow line) with normalized complex amplitudes. a** $A_{f_1} = e^{i\pi/4}$, $A_{f_2} = e^{-i\pi/4}$; **b** $A_{f_1} = \frac{1}{2}e^{i\pi/4}$, $A_{f_2} = e^{i\pi/2}$; **c** $A_{f_1} = \frac{1}{2}e^{-i\pi/4}$, $A_{f_2} = \frac{1}{2}e^{i\pi/2}$. **d–f** The corresponding time-varying reflection coefficients of Partitions 1# and 2# for the manipulations of $f_1$ and $f_2$, respectively. The frequency of incident waves is 4.25 GHz, and the modulation frequencies of the two regions are $\triangle f_1 = 100$ kHz and $\triangle f_2 = 200$ kHz, respectively. The generated harmonic orders are −1st and +1st in Partitions 1# and 2#, so that the corresponding frequencies are $f_1 = 4.2499$ GHz and $f_2 = 4.2502$ GHz.

and +44°, respectively. To validate the calculations, the far-field scattering patterns were measured in the microwave anechoic chamber. Figure 3a and b display the results of dual partitions operated alternatively, while Fig. 3c and d display the results of dual partitions operated simultaneously. It can be seen that the interference does exist when the dual partitions are modulated simultaneously, which results in slight deviations of the original scattering patterns at different bands. But the influence is limited. On the other hand, when the dark green and orange columns are arranged alternatively, the scattering beams at the frequencies $f_1$ and $f_2$ are directed to −21° and + 21°, respectively, which can be found in Supplementary Fig. S2 in Supplementary Information.

We need to stress that this concept can be further extended to the case of multiple harmonic generation and manipulation. In fact, the whole metasurface can be divided into more partitions for controlling more harmonics. In theory, each meta-atom can be regarded as an electrically small antenna that can generate dynamic amplitude and phase responses required by harmonic modulation under the illumination of RF signals, and then radiates the energy back into free space. In addition, the shapes of partitions can be elaborately designed to meet the demand of wavefront shaping for various harmonic frequencies. This is critical for communication applications since each harmonic can serve as an individual carrier to send the information to the user end at various locations. Figure 4 gives the ideal simulations of multi-frequency scattering patterns when all partitions are simultaneously modulated under the incident frequency of 4.25 GHz. As shown in Fig. 4a, the metasurface is divided into four partitions with the modulation frequencies of $\Delta f_1 = 100$ kHz, $\Delta f_2 = 200$ kHz, $\Delta f_3 = 300$ kHz, and $\Delta f_4 = 400$ kHz. At first, the main lobes of the scattering patterns of the four partitions are respectively designed to point to −60°, −30°, 0°, and 50° by independently controlling the phase gradients of the four partitions. Since different modulation frequencies are applied to the four partitions, the scattering patterns can be individually manipulated.

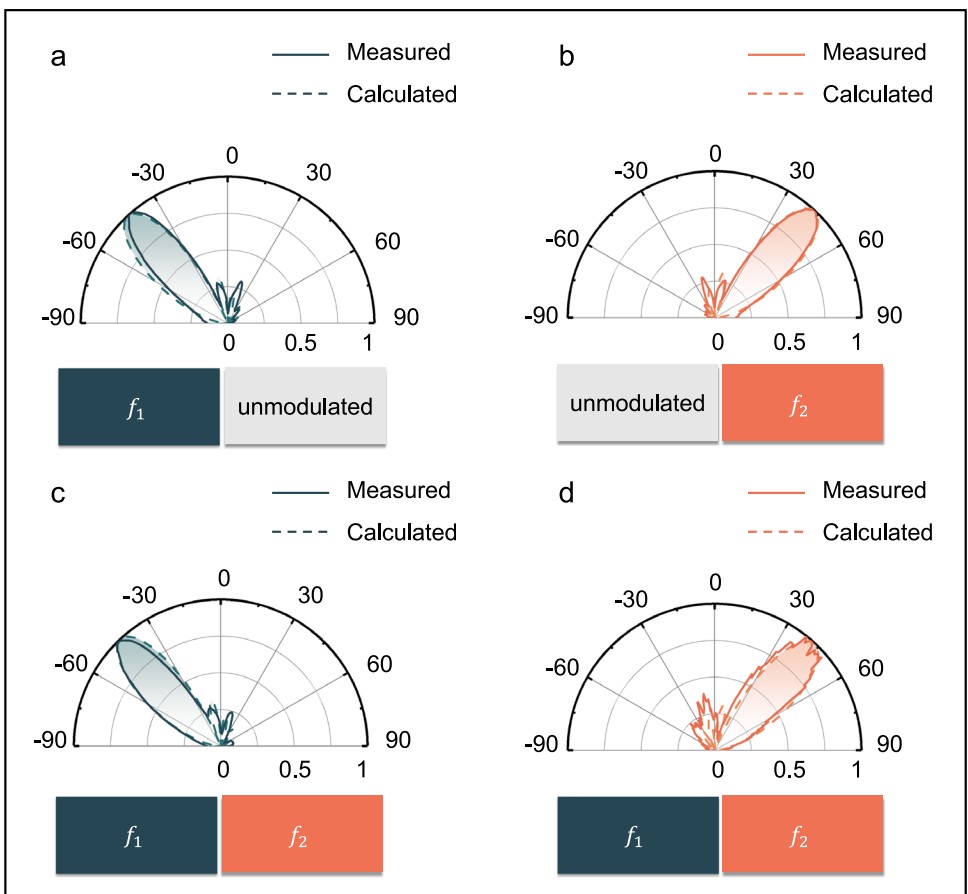

**Fig. 3 | The measured and calculated scattering patterns from $f_1$ (dark green) and $f_2$ (orange) partitions. a, b** The left and right partitions are modulated alternatively. **c, d** The left and right partitions are modulated simultaneously.

To demonstrate the fact that the far-field beams generated by different harmonics are independent of each other, in Fig. 4b, by revising the phase gradient in the $f_3$ partition, we can change the main-lobe angle from 0° to 17°. Moreover, by changing the phase gradient of the $f_1$ partition, the main-lobe angle can be changed from −60° to −44°, as shown in Fig. 4c. In fact, the metasurface can be further divided into eight partitions by using the modulation frequencies $\Delta f_1 = 100$ kHz, $\Delta f_2 = 200$ kHz, $\Delta f_3 = 300$ kHz, $\Delta f_4 = 400$ kHz, $\Delta f_5 = 500$ kHz, $\Delta f_6 = 600$ kHz, $\Delta f_7 = 700$ kHz, and $\Delta f_8 = 800$ kHz. Initially, the main lobes of the scattering patterns of the eight partitions are pointed at −60°, −50°, −30°, −10°, 0°, 20°, 40°, and 60°, which are illustrated in Fig. 4d. Similarly, as shown in Fig. 4e, to independently steer the main lobe of the $f_5$ partition from 0° to 7°, only the phase gradient of the $f_5$ partition needs to be reset. As shown in Fig. 4f, to independently steer the main lobe of the $f_1$ partition from −60° to −65°, only the phase gradient of the $f_1$ partition needs to be revised.

**Frequency-division multiplexing transmitter based on ASTCM**
In order to demonstrate the strong capabilities of multiple frequency manipulations, we have established a frequency-division multiplexing (FDM)[43,44] wireless transmitter with the aid of ASTCM. In this transmitter, ASTCM plays core roles in two aspects: carrier-wave generation and signal modulation. A number of carrier frequencies are produced under the excitation of monochromatic wave during the wave-matter interactions, and each of them is modulated by Quadrature Phase Shift Keying (QPSK) signal. The designed ASTCM can be found in Fig. 5a, which is composed of 8 × 16 meta-atoms, as shown in Fig. 5b. From Fig. 5c, under different biasing voltages, the measured reflection amplitudes are greater than −3 dB, indicating the presence of material

losses near the resonance frequency. Note that the measured reflection phase range is beyond 2π. The achieved full phase range is especially important to realize a high conversion efficiency from the fundamental frequency to the first order harmonic[33]. The reflectivity of the meta-atoms in wide bandwidth can be found in Supplementary Fig. S3 of Supplementary Information.

To confirm the ability of dual-harmonic generations with the proposed scheme, a transmitting horn antenna is used to excite the ASTCM sample, as shown in Fig. 5a. The ASTCM sample is controlled by a commercial platform (PXIe-1082, NI Corp.). A pair of receiving antennas are employed to receive the signals reflected from the ASTCM at different carrier frequencies.

As shown in Fig. 6a, two middle columns (in red and black colors) of the metasurface are modulated with the time-varying biasing signals, while the remaining white columns are unmodulated. The modulation signals are used to keep the reflection phases varying from 0 to 2π continuously in the period of 5 us and 10 us for the red and black column, respectively. Thus, the incident frequency can be shifted to the first-order harmonic with high efficiency at 4.2502 GHz at the interface of the two columns[34]. The signals from ASTCM are monitored by a software defined radio reconfigurable device (USRP-2943R, National Instruments Corp), and mixed with a local oscillator signal at 4.25 GHz. In consequence, we can obtain the measured baseband spectra, as illustrated in Fig. 6b. It is obvious that there are two peaks at 100 kHz and 200 kHz, indicating that the two frequency signals are successfully synthesized using ASTCM as expected. We also find a large DC signal in the spectra, which is originated from the carrier signal reflected from the unmodulated columns. Here we note that the high-order harmonic magnitudes are close to that of the fundamental

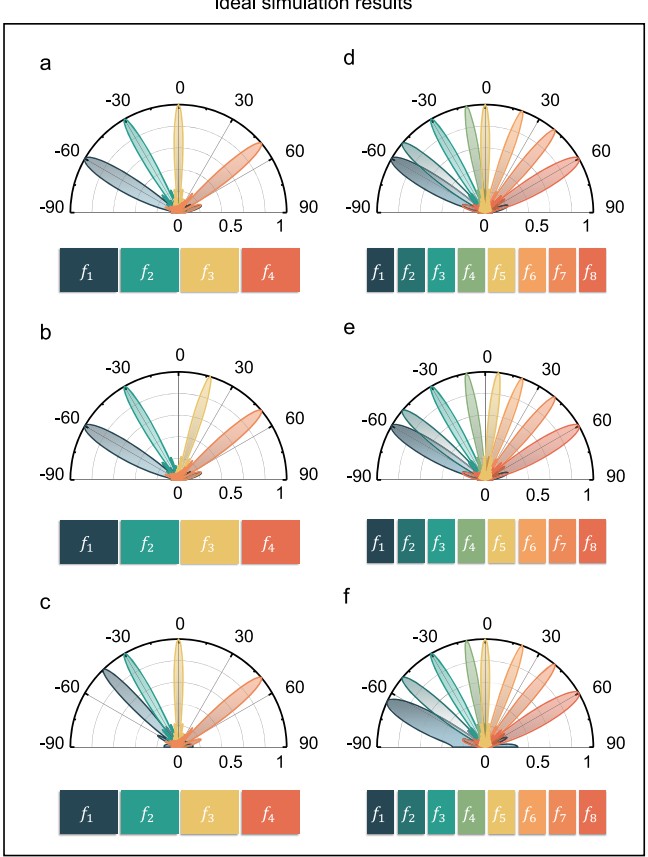

Ideal simulation results

**Fig. 4 | Ideal simulation scattering patterns at multiple frequencies generated by ASTCM when all partitions are modulated simultaneously. a–c** The metasurface is divided into four partitions with the modulation frequencies $\Delta f_1 = 100$ kHz, $\Delta f_2 = 200$ kHz, $\Delta f_3 = 300$ kHz, and $\Delta f_4 = 400$ kHz. **d–f** The metasurface is divided into eight partitions with the modulation frequencies $\Delta f_1 = 100$ kHz, $\Delta f_2 = 200$ kHz, $\Delta f_3 = 300$ kHz, $\Delta f_4 = 400$ kHz, $\Delta f_5 = 500$ kHz, $\Delta f_6 = 600$ kHz, $\Delta f_7 = 700$ kHz, and $\Delta f_8 = 800$ kHz. The main lobes of the four-partition case and eight-partition case are directed at different angles. **a** $f_1 \sim f_4$: −60°, −30°, 0°, 50°; **b** $f_1 \sim f_4$: −60°, −30°, 17°, 50°; **c** $f_1 \sim f_4$: −44°, −30°, 0°, 50°. **d** $f_1 \sim f_8$: −60°, −50°, −30°, −10°, 0°, 20°, 40°, 60°; **e** $f_1 \sim f_8$: −60°, −50°, −30°, −10°, 7°, 20°, 40°, 60°; **f** $f_1 \sim f_8$: −65°, −50°, −30°, −10°, 0°, 20°, 40°, 60°.

harmonic, owing to the spatial multipath and shading effects when the two antennas and sample are closely deployed, leading to large fluctuation of the harmonic energy. Some high-order harmonics emerge as well due to the fact that the linearity of the phase curve of the metaatom is not perfect, resulting in the undesired harmonics in experiments.

Similarly, when more columns are modulated in the experiments, we expect to get more frequency signals during the wave reflections. In Fig. 7a, there are 16 columns in ASTCM, and we divide it into eight regions, which means that two adjacent columns share the same modulation frequency. The modulation frequencies of the eight regions from the left to right are 400 kHz, 500 kHz, 600 kHz, 700 kHz, 150 kHz, 350 kHz, 550 kHz, and 850 kHz, respectively. Due to the limited number of the analog-to-digital converter (ADC) channels, we can only provide 4 biasing voltages simultaneously. For the purpose of generating eight frequency signals, all columns are separated into two groups and modulated alternatively. To increase the guard bands among the multi-frequency signals, we convert the incident wave to the −1st order harmonic in the left four partitions and the +1st order harmonic in the right four partitions. The corresponding baseband spectra are plotted in Fig. 6c, d. It is clear that the eight frequencies are well produced by the proposed method. In addition, owing to the

bandwidth limitation of metasurface, the nonlinearity of controlling module, and the sightly unequal distances from the receiving antenna to different meta-columns, the reflected energies received at different harmonics are not strictly equal.

Next, we exploit the eight frequencies generated by the ASTCM as the carrier frequencies to transmit eight channels of QPSK signals. To realize the directed phase modulations at the radio frequencies, all columns should have four reflective phase states that correspond to four baseband symbols. Here, we introduce four kinds of time delays $[T_0/8, 3T_0/8, 5T_0/8, 7T_0/8]$ to the temporal reflectivity, in which $T_0$ represents the corresponding modulation period. The time delays lead to the phase lags of 0°, 90°, 180°, and 270° from Eq. (6). In addition, owing to the restriction of the carrier frequency interval, the symbol rates of all channels are set to 50 kB/s. Due to the limited ADC channels as stated above, the right half of the metasurface is firstly used for transmitting four data streams, and then the left half is activated for transmitting another four data streams, as shown in Fig. 7a. Since USRP only supports two receiving channels, two horn antennas are employed in Fig. 5a to record two data streams at the same time. A time-division strategy is adopted to receive the eight data streams in turn, and in each time slot the information is received at two selected carrier frequencies. Eight pictures illustrated in Fig. 7b are modulated at the synthesized frequencies, and the modulation signals are transmitted from the metasurface to USRP. The measured signal-to-noise ratios (SNRs) can be found in Note 4 of Supplementary Information. A video on the multi-channel wireless communication is provided in Supplementary Movie 1.

The constellation diagrams at the eight carrier frequencies are measured, as illustrated in Fig. 8a–h (left panels). The four constellation points can be well distinguished, which ensures low bit error rate in the picture transmissions, as illustrated in the right panels of Fig. 8a–h. We note that in Fig. 8a–h, the scatter sizes at some frequencies are relatively large, as can be ascribed to the energy fluctuation among multiple frequencies. This is related to the spatial multipath and shading effects among the sample and antennas as mentioned above. In these cases, however, the quality of transmitted pictures is still very good, as shown in the right panels of Fig. 8d–h. To further improve the performance of the ASTCM-based transmitter in the future, an ADC with higher sampling rate and an ASTCM with broader bandwidth need to be explored. It is worth noting that with the increase of modulation frequency, it is also feasible to construct a wireless receiver with the proposed ASTCM, as discussed in Supplementary Information Note 5.

Usually the transmission directions of the eight channels are along the specular reflection angles related to the normal direction when the partitions have uniform space coding. However, they can be designed independently with predefined deflection angles. Next we will prove that ASTCM can transmit two distinct data streams simultaneously at dual frequencies to different deflection angles by experiments. The ASTCM is equally divided into left and right partitions by modulating with two frequencies $\Delta f_1 = 200$ kHz and $\Delta f_2 = 500$ kHz. The operating frequency is 4.25 GHz. By changing the phase gradients of space coding in each partition, the main lobe of the corresponding scattering pattern can be directed to the desired angle. Here we design two sets of scattering patterns, whose main lobes are directed to around 0° and −44°, respectively.

The experimental scenario is illustrated in Fig. 9. Specifically, the ASTCM is illuminated by a transmitting horn antenna at 4.25 GHz, which is connected to a microwave signal generator (Keysight E8267D). The ASTCM is modulated with a controlling platform (PXIe-1082, NI Corp.) consisting of a high-speed I/O bus controller, an FPGA module, a digital-analog conversion module, a DC power supply module, and a timing module, which can provide high-quality biasing signals on the embedded varactor diodes within the metasurface. In the receiving end, two receiving antennas (Receivers A and B) are

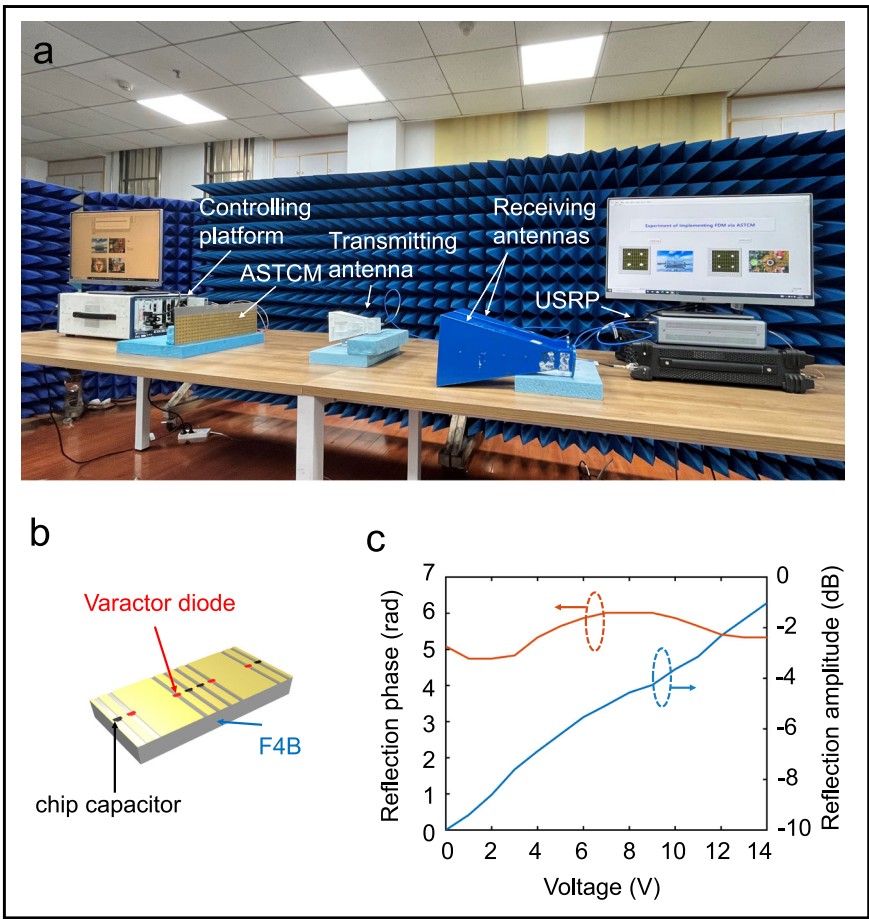

**Fig. 5 | Details of the experiment configuration and element design. a** Experimental configuration of the multi-frequency generations and multi-channel signal transmissions. **b** Structure of the meta-atom. **c** Measured amplitude (red) and phase spectra (blue) of the metasurface as the external bias voltage changes.

employed to record the echo waves from the metasurface with a software-defined radio transceiver (NI USRP RIO 2943R). Receiver A is located in the direction normal to the ASTCM, while Receiver B is deviated −44° from the normal direction. The distance between ASTCM and the receiving antennas is 3 m.

We investigate three cases illustrated in Fig. 10a, d, and g, which have different main-lobe angles at 4.2502 GHz and 4.2505 GHz. During experiments, the two partitions of ASTCM are enabled simultaneously. In Case 1 shown in Fig. 10a, the main lobes of the scattering patterns are directed to 0° at both 4.2502 GHz and 4.2505 GHz. Receivers A and B are located at 0° and −44°, respectively. As illustrated in Fig. 10b, the constellations from Receiver A at the two frequencies are distinguishable. In contrast, the quality of constellations at 4.2502 GHz and 4.2505 GHz in Receiver B greatly deteriorates owing to the deviation angle. Only the RF energy from the sidelobe of scattering waves of the metasurface can be obtained by Receiver B, thereby the corresponding recovered pictures are blurred, as depicted in Fig. 10c.

In Case 2 illustrated in Fig. 10d, the main-lobe angle is 0° at 4.2502 GHz and is −44° at 4.2505 GHz, while Receivers A and B are located at 0° and −44°, respectively. Figure 10e shows that the constellation from Receiver A at 4.2502 GHz has high quality, but it is poor at 4.2505 GHz in the presence of the deviation angle between Receiver A and the scattering direction at 4.2505 GHz. Similarly, Fig. 10f presents a poor constellation from Receiver B at 4.2502 GHz and a good constellation at 4.2505 GHz. In Case 3 shown in Fig. 10g, the main lobe angles are −44° at 4.2502 GHz and 0° at 4.2505 GHz, which is opposite to Case 2. Receivers A and B are located at 0° and −44°, respectively. Figure 10h demonstrates that the constellation from Receiver A at 4.2502 GHz is blurred, but is distinguishable at 4.2505 GHz, as

expected. On the contrary, Fig. 10i illustrates that the recovered picture and constellation from Receiver B are good at 4.2502 GHz but poor at 4.2505 GHz.

## Discussion

A strategy of multi-frequency generations and harmonic beam manipulations is proposed by using a multi-partition ASTCM. Different from the previous approaches that rely on the complicated coding algorithms for synchronous STC digital metasurface, the proposed multi-partition ASTCM can enhance the number of independently controlled high-order harmonics and simplify the coding strategy with high nonlinear conversion efficiency. To prove the nonlinear manipulation capability of the proposed multi-partition ASTCM, experiments are conducted where eight carrier frequencies are produced by the ASTCM and are used to transmit eight data streams of different pictures. The presented work paves a way for new architectures of frequency-division multiplexing wireless communication systems.

## Methods

### Details on the asynchronous space-time-coding digital metasurface

ASTCM used in this work operates at the central frequency of 4.25 GHz, and contains 8 × 16 meta-atoms. It is designed by the commercial software, CST Microwave Studio 2016, and fabricated with printed circuit board technology. Each meta-atom is a three-layered structure, in which the top and bottom surfaces are made of copper, the middle substrate is F4B ($\varepsilon_r = 3.0$, tan$\sigma = 0.0015$) with a thickness of 5 mm. In addition, four varactor diodes (SMV-2019, Skyworks, Inc.) and four chip capacitors (0.1 pF) are soldered across

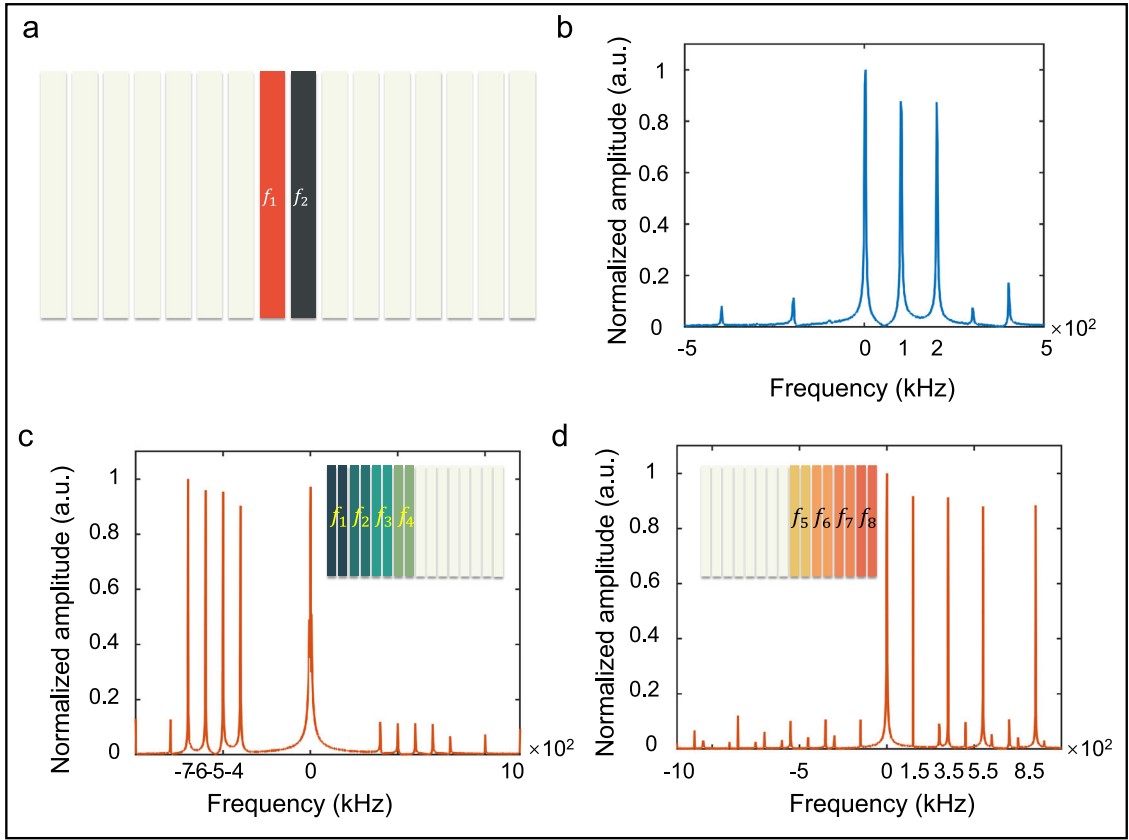

**Fig. 6 | Frequency generation with part of the metasurface columns.**
**a** Generation of dual frequencies ($f_1$, $f_2$) with two columns (red and black) in the metasurface, while the remaining white columns are unmodulated. **b** Baseband spectra for dual-frequency generation. The abbreviation "a. u." is for "arbitrary units". **c, d** Baseband spectra for eight-frequency generations ($f_1$-$f_8$), where the left and right eight columns in the metasurface are responsible for synthesizing the four frequencies, respectively.

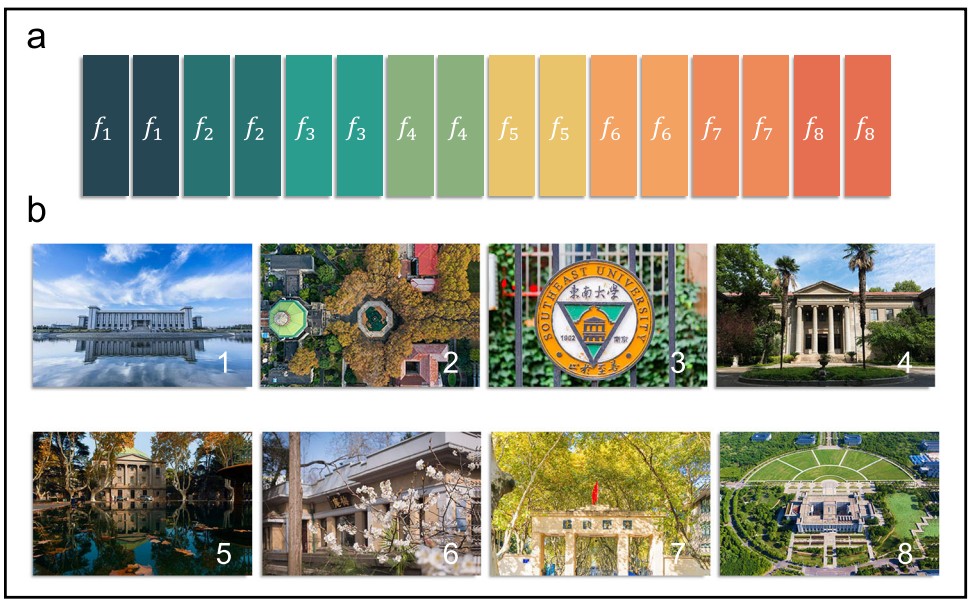

**Fig. 7 | Pictures to be transmitted by different metasurface columns modulated with different carrier frequencies. a** Space division of the metasurface for eight-frequency synthesizing, in which two neighboring columns with the same color share the identical modulation frequency. **b** Eight pictures to be transmitted at the carrier frequencies ($f_1$-$f_8$) of 4.2496 GHz, 4.2495 GHz, 4.2494 GHz, 4.2493 GHz, 4.25015 GHz, 4.25035 GHz, 4.25055 GHz, and 4.25085 GHz.

**a** Region 1#
4.2496 GHz

**b** Region 2#
4.2495 GHz

**c** Region 3#
4.2494 GHz

**d** Region 4#
4.2493 GHz

**e** Region 5#
4.25015 GHz

**f** Region 6#
4.25035 GHz

**g** Region 7#
4.25055 GHz

**h** Region 8#
4.25085 GHz

**Fig. 8 | Constellation diagrams and corresponding recovered pictures at different carrier frequencies of the frequency-division multiplexing wireless communication system. a–h** 4.2496 GHz, 4.2495 GHz, 4.2494 GHz, 4.2493 GHz, 4.25015 GHz, 4.25035 GHz, 4.25055 GHz, and 4.25085 GHz.

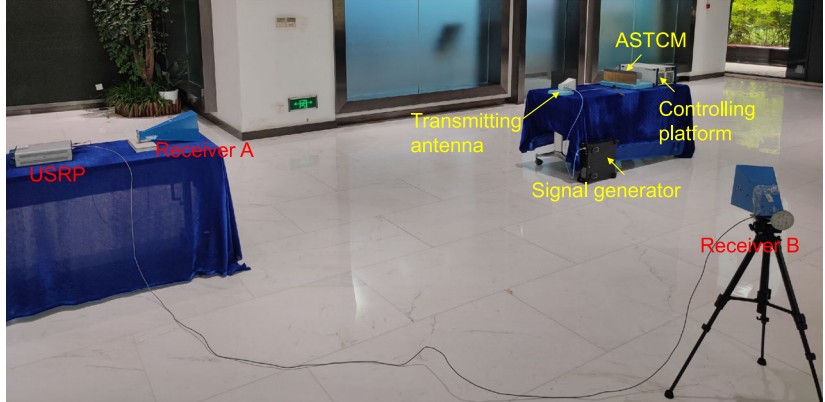

**Fig. 9 | Experimental scenario of data transmissions at two frequencies toward different deflection angles.** The ASTCM is illuminated by a transmitting horn antenna at 4.25 GHz, which is connected to a signal generator. The ASTCM is modulated with a controlling platform. Two receiving antennas (Receivers A and B) connected to a USRP are employed to record the echo waves from the metasurface.

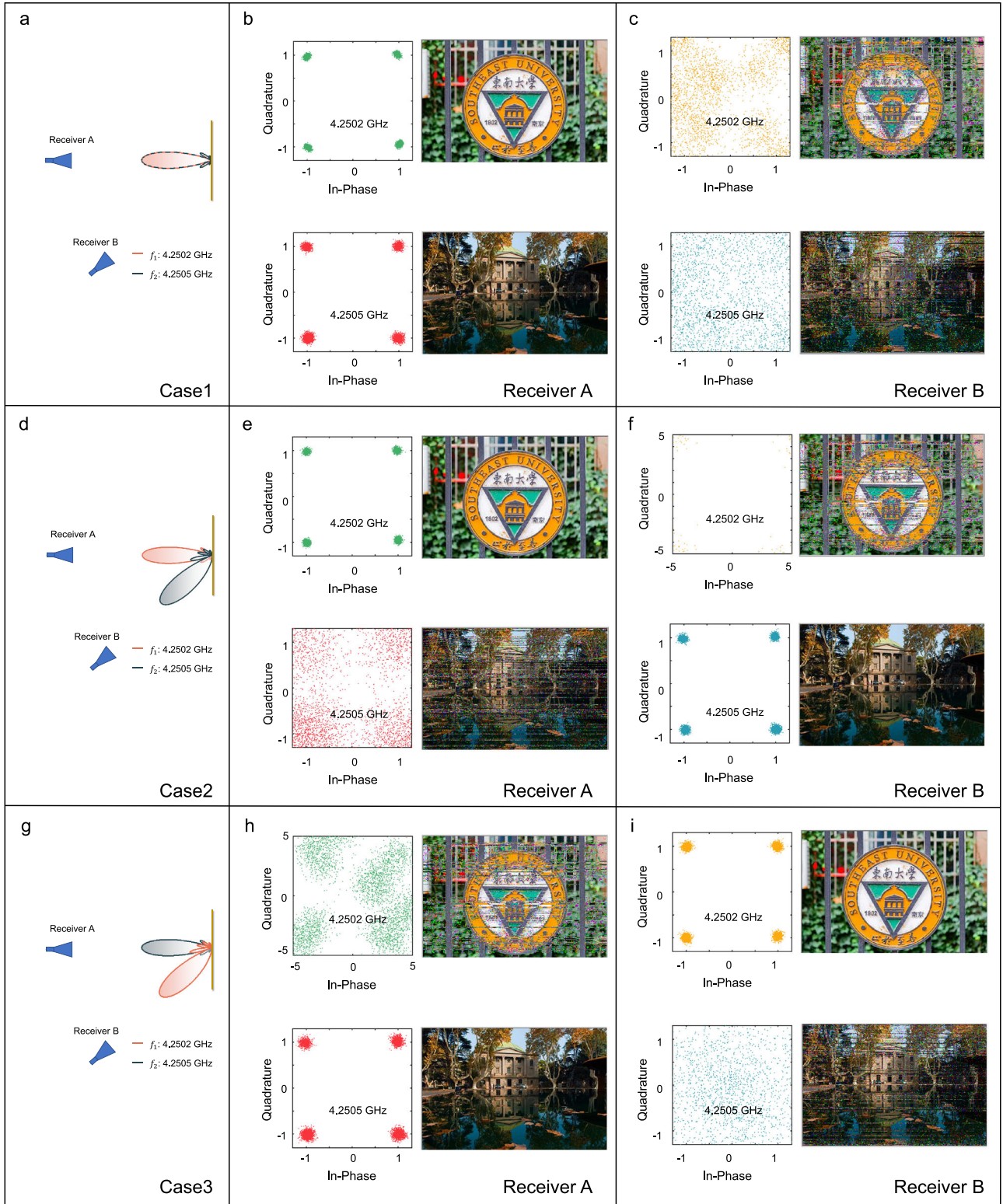

**Fig. 10 | Measured constellations and recovered pictures at the frequencies of 4.2502 GHz and 4.2505 GHz in three cases. a** Case 1: the main lobes of the scattering patterns from $f_1$ and $f_2$ partitions are both directed to 0°. **b** Case 1: Results in Receiver A. **c** Case 1: Results in Receiver B. **d** Case 2: the main lobes of the scattering patterns from $f_1$ and $f_2$ partitions are directed at 0° and −44°, respectively. **e** Case 2: Results in Receiver A. **f** Case 2: Results in Receiver B. **g** Case 3: the main lobes of the scattering patterns from $f_1$ and $f_2$ partitions are directed at −44° and 0°, respectively. **h** Case 3: Results in Receiver A. **i** Case 3: Results in Receiver B.

the metallic slots on the top of the meta-atom for altering its reflection phase with different biasing voltages. In the simulations, the diode can be modeled as a capacitor-resistor series circuit as suggested in refs. 33,34.

## Experimental setup
The experimental configuration is illustrated in Fig. 5a for confirming the ability of harmonic generations and manipulations with the proposed scheme. A linearly polarized transmitting horn antenna is used

to excite the ASTCM sample. The ASTCM sample is controlled by a commercial platform (PXIe-1082, NI Corp.) consisting of a high-speed I/O bus controller, an FPGA module, a digital-analog conversion module, a DC power supply module, and a timing module, which enables us to provide high-quality biasing signals on the embedded varactor diodes in the metasurface. A microwave signal generator (Keysight E8267D) is connected to the antenna and provides a single frequency signal at 4.25 GHz. Two receiving antennas are placed behind the transmitting antenna to record echo waves from the metasurface with a software-defined radio transceiver (NI USRP RIO 2943 R), and receive two data streams during the wireless communication experiments. The distance between the metasurface and receiving antennas is 1.2 m.

## Data availability

The authors declare that all relevant data are available in the paper and its Supplementary Information files, or from the corresponding author on request.

## Code availability

The custom computer codes used in this study are available from the corresponding authors on request.

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

## Acknowledgements

This work was supported by the National Natural Science Foundation of China (62288101, 61731010 and 62201139), the National Key Research and Development Program of China (2018YFA0701904, 2017YFA0700201, 2017YFA0700202, 2017YFA0700203), the Jiangsu Province Frontier Leading Technology Basic Research Project (BK20212002), Fundamental Research Funds for the Central Universities (2242022k30004, 2242022R10185), the National Science Foundation (NSFC) for Distinguished Young Scholars of China (62225108), the 111 Project (111-2-05), the Program of Song Shan Laboratory (Included in the management of Major Science and Technology Program of Henan Province under Grants 221100211300-02 and 221100211300-03), and the Southeast University - China Mobile Research Institute Joint Innovation Center (R207010101125D9).

## Author contributions

T.J.C. suggested the designs and planned and supervised the work, in consultation with Q.C., S.R.W. and J.Y.D. conceived the idea and carried out the theoretical analysis and numerical simulations. T.J.C., Q.C., S.R.W., and J.Y.D., wrote the manuscript. S.R.W., J.Y.D., Q.Y.Z., J.C.K., Q.C. and T.J.C. participated in the experiments and reviewed the manuscript.

## Competing interests

The authors declare no competing interests.
