## [Peer Review File · Nature Communications]

REVIEWER COMMENTS

Reviewer #1 (Remarks to the Author):

In this paper, a multi-partition asynchronous space-time-coding digital metasurface (ASTCM) is proposed for manipulating multiple frequencies simultaneously. As an application, it is used to implement wireless communications with frequency division multiplexing. The multi-partition ASTCM makes a big step compared to previously reported digital coding metasurface, since it can directly modulate the incident electromagnetic waves with multiple data streams freely with the restriction of phase bit number. In general, it is an interesting work. I think the following issues should be carefully addressed before I recommend it for publication:

1. The core of the multi-partition ASTCM is to make different regions of a single metasurface independently work for different functions, such as beamforming, wireless transmission, etc. But these functions can also be completed with multiple metasurfaces. The authors should explain the advantages of the proposed multi-partition ASTCM.
2. According to Eq. (7), each demarcated partition can manipulate one selected harmonic by designing a temporal reflection phase. Please give the conversion efficiency of the harmonic and discuss how to improve harmonic purity.
3. In Fig. 5c, the authors only displayed the reflectivity of the designed meta-atom at 4.25 GHz. But in practice the ASTCM was used to transmit pictures at multi-frequency channels. The author should discuss the reflection properties of the ASTCM at other frequencies.
4. In Fig. 6c and 6d, the energies of generated multiple frequencies are not equal. The authors should give detailed explanations on that point.
5. How about the working bandwidth of this ASTCM, since it is connected with the available data transmission rate.
6. From Fig. 6c, I guess that the left four partitions were used to generate the -1st-order harmonics. But there are no descriptions about that.

Reviewer #2 (Remarks to the Author):

In this paper, a strategy of multi-frequency generations and harmonic beam manipulation is proposed. Specifically, a digital metasurface is spatially divided into several parts to excite different harmonics and transmit different pictures independently in each partition. Compared with previous approach, the proposed strategy can enhance the number of independently controlled high-order harmonic-channels without complicated coding algorithms. Experiment results demonstrate that eight harmonic-channels are produced by the ASTCM and are applied to transmit eight data streams successfully. However, in traditional wireless communication, both information emitting and receiving functions are always required simultaneously in either base stations (or microcell) or terminal devices, while the reported ASTCM can only work for transmitting signals. It would be more complicated and more difficult to maintain if different RF systems are used in one wireless communication system. Another concern is that the signal modulation speed is limited to few hundreds of kHz by the diode used in the device which will restrict communication performance. So the authors need to explain clearly in what kind of real world application scenarios the proposed device could be useful. Besides, I have other issues with the present submission that the authors need to consider, which are listed in the following:

1. As illustrated in Fig. 2, the modulation frequencies of the two partitions are $\Delta f_1=100$ kHz and $\Delta f_2=200$ kHz, however, the duration (defined as $1/\Delta f$) of Partition 2# is clearly larger than that of Partition 1# in Fig. 2(d)-(f). Is there any problem with the corresponding time-varying reflection coefficients for different frequency components? Please explain.
2. The scale values of the reflection phase in Fig. 5(c) are not identical with the description of “Note that the reflection phase range is beyond 2π .” The authors should check the figures carefully.
3. According to the article, a single column of the proposed ASTCM can be modulated to generate a harmonic signal. However, in general, subwavelength elements should be periodically arranged to a certain size, e.g., several wavelengths, to realize the ideal modulation of electromagnetic waves. Is there any theoretical support for using a unitary structure of less than half a wavelength to achieve precise harmonic modulation? In addition, the adjacent columns are loaded with different modulations to generate harmonics of different frequencies. With such a small interval between adjacent columns, will there be frequency mixing and phase distortion between the generated harmonics? Please explain the problems in detail.
4. The independence of the far-field patterns generated on each harmonic is crucial for the communication channels. Fig. 3 and Fig. 4 show the far-field patterns of different harmonic modulations. However, only one case is given as a demonstration for each scenario of different harmonic combinations. The authors should provide more cases to demonstrate that the far-field beams generated by each harmonic are independent of each other when the metasurface is divided into several partitions.

5. The measured results of the far-field patterns on different harmonics should be given to validate the proposed strategy.

6. Are the eight channels applied to transmit pictures in experiment established at different deflection angles on each harmonic or at the specular reflection angles of different harmonics? The authors should clarify it in the article. If the channels of eight harmonics are established at different deflection angles, please add the measured results of far-field patterns of the different harmonics to demonstrate the feasibility of QPSK. If the channel of each harmonic is established at the angle of specular reflection, how to transmit eight different data streams at deflected angles? And is it possible to transmit different pictures in eight deflected-angle channels with QPSK modulation?

Reviewer #3 (Remarks to the Author):

The submitted manuscript presents an asynchronously modulated metasurface for frequency division multiplexing of information. The presented results are interesting and can be of interest across different disciplines. I believe the following (in order of appearance) need to be addressed before this paper is acceptable for publication:

1) In the current format, the manuscript shares too much similarity with reference 42. One may argue that up to 2.2 (page 12), the material is presented in Ref 42 as well. Please provide a clarification on the novelty and contribution of this work over 42.

2) On page 2, the authors refer to "light-matter" interaction, while this work uses RF signals. Please reword.

3) The phrase "the quantification bit number of the reflection phase" on page 3 is not clear. Before that, no mention of quantification but number, reflection phase are presented. Please reword.

4) What is Gamma in equation 4?

5) On page 7, the authors state that we would need a large number of PIN diodes to achieve desired phase states. If instead of PIN diodes, authors had used varactor diodes, would that be solved?

6) The radiation patterns in figure 3 and 4 are presented without any scale. As a result, we cannot deduce the level of sidelobes. Please fix that.

7) Can this device steer the reflected beam toward arbitrary angles or specific angles? The results in fig 4 are presented for increments of 10 degrees.

8) In figure 5c, the range of reflection angles are only 6 degrees. Is that intended? Please elaborate on how the amplitude and phase been measured.

9) What is the interference between different bands? It seems the patterns generated in figure 4 should result in interference between different bands.

10) When we partition the aperture the way the authors suggest, we also reduce the directivity. Does that reduce the SNR? If yes, have you measured it? This is an important observation. Because now we have a device that has lower SNR and increased interference (see previous comment). So these observations and studies are important to compare this device with traditional counterparts.

Responses to Referees' Comments

The authors appreciate the constructive comments and suggestions from all referees, which help improve the quality of the manuscript and make the manuscript more complete. We revised the original manuscript carefully based on these comments, and the revised parts are marked with **red fonts**. Below are our replies (**in blue fonts**) to the comments (in black fonts).

To Referee #1:

Comment:

In this paper, a multi-partition asynchronous space-time-coding digital metasurface (ASTCM) is proposed for manipulating multiple frequencies simultaneously. As an application, it is used to implement wireless communications with frequency division multiplexing. The multi-partition ASTCM makes a big step compared to previously reported digital coding metasurface, since it can directly modulate the incident electromagnetic waves with multiple data streams freely with the restriction of phase bit number. In general, it is an interesting work. I think the following issues should be carefully addressed before I recommend it for publication.

Response:

Thank you very much for your positive comments. We have addressed all your concerns in the following detailed responses.

Comment:

Q1. The core of the multi-partition ASTCM is to make different regions of a single metasurface independently work for different functions, such as beamforming, wireless transmission, etc. But these functions can also be completed with multiple metasurfaces. The authors should explain the advantages of the proposed multi-partition ASTCM.

Response:

Thank you for your good question. The proposed multi-partitioned ASTCM offers a low-cost and simplified platform to achieve multiple functionalities compared to multiple metasurfaces. The sizes and shapes of the working partitions can be dynamically adjusted according to the requirements of different tasks.

Comment:

Q2. According to Eq. (7), each demarcated partition can manipulate one selected harmonic by designing a temporal reflection phase. Please give the conversion efficiency of the harmonic and discuss how to improve harmonic purity.

Response:

Thank you for your valuable comment. Theoretically, the harmonic conversion efficiency depends on the modulated phase waveform of the reflection coefficient, which follows by (see Ref. [S1]):

$$\varphi(t) = p \times \text{mod}(t, T) \quad (\text{S1})$$

in which p is the phase slope, $\text{mod}(t, T)$ denotes the remainder of t / T , and T represents the modulation period of the metasurface. For the m^{th} order harmonic, we have $pT = 2m\pi$. In general, the continuous reflection phase is discretized into finite phase states with the phase step of $2\pi/2^n$, where n is the quantization bit number. The conversion efficiency is defined as the ratio of the energy of the generated harmonics to the total energy of the incident waves ^[S1].

From Ref. [S1], it is clear that the harmonic purity tends to be improved with the increase of the quantization bit number n .

[S1]. J. Y. Dai, *et al.*, High-Efficiency Synthesizer for Spatial Waves Based on Space-Time-Coding Digital Metasurface. *Laser & Photonics Reviews* **14**(6), 1900133, 2020.

Comment:

Q3. In Fig. 5c, the authors only displayed the reflectivity of the designed meta-atom at 4.25 GHz. But in practice the ASTCM was used to transmit pictures at multi-frequency channels. The author should discuss the reflection properties of the ASTCM at other frequencies.

Response:

Thank you for your good question. As illustrated in Figs. R1(a) and (b) below, we provide the simulated reflection amplitude and phase spectra of the ASTCM element ranging from 2GHz to 6 GHz based on CST Microwave Studio 2016. It can be seen that the reflection amplitudes under different biasing voltages are greater than -4 dB while the reflection phase range is beyond 2π within (4.25 ± 0.1) GHz. The simulation results are also added in the **Supplementary Information**.

Fig. R1. Reflection amplitude and phase spectra of the meta-atom at different biasing voltages.

Comment:

Q4. In Fig. 6c and 6d, the energies of generated multiple frequencies are not equal. The authors should give detailed explanations on that point.

Response:

Thank you for your valuable comment. There are two reasons that the energies of generated multiple frequencies are not equal. On the one hand, due to the bandwidth limitation of the metasurface and the nonlinearity of controlling module, the harmonic conversion efficiencies are different with the change of the modulation frequency. On the other hand, as the distances between the metasurface columns and the receiving antenna are slightly different, the received reflected energies are not strictly equal. These discussions are added to line 12, paragraph 2, page 16 of the revised paper.

Comment:

Q5. How about the working bandwidth of this ASTCM, since it is connected with the available data transmission rate.

Response:

Thank you for your good question. The working bandwidth of this ASTCM is about 200 MHz, which can be found in Fig. R2 below. In order to increase the data transmission rate, we will try to further expand the bandwidth of the metasurface by updating the element design.

Fig. R2. Reflection amplitude and phase spectra of the meta-atom at different biasing voltages.

Comment:

Q6. From Fig. 6c, I guess that the left four partitions were used to generate the -1st-order harmonics. But there are no descriptions about that.

Response:

Thank you very much for your valuable comment. We added the descriptions as suggested in Line 8, paragraph 2, page 16 in the revised manuscript as follows:

“To increase the guard bands among the multi-frequency signals, we convert the incident wave to the -1st order harmonic in the left four partitions and the +1st order harmonic in the right four partitions.”

To Referee #2:

Comment:

In this paper, a strategy of multi-frequency generations and harmonic beam manipulation is proposed. Specifically, a digital metasurface is spatially divided into several parts to excite different harmonics and transmit different pictures independently in each partition. Compared with previous approach, the proposed strategy can enhance the number of independently controlled high-order harmonic-channels without complicated coding algorithms. Experiment results demonstrate that eight harmonic-channels are produced by the ASTCM and are applied to transmit eight data streams successfully.

However, in traditional wireless communication, both information emitting and receiving functions are always required simultaneously in either base stations (or microcell) or terminal devices, while the reported ASTCM can only work for transmitting signals. It would be more complicated and more difficult to maintain if different RF systems are used in one wireless communication system.

Another concern is that the signal modulation speed is limited to few hundreds of kHz by the diode used in the device which will restrict communication performance. So the authors need to explain clearly in what kind of real world application scenarios the proposed device could be useful. Besides, I have other issues with the present submission that the authors need to consider, which are listed in the following:

Response:

Thank you very much for your constructive comments and suggestions.

For the first concern, the ASTCM in this work is still used for the transmitter. However, it can also be employed as a receiver. The received signals from different columns of ASTCM can be modulated by orthogonal codes, then they will be received by a horn antenna connected by a signal channel receiver, and separated at the baseband by convolution operation with the same orthogonal codes. We will continue to work on the new system chain, so that both information emitting and receiving functions can be simultaneously implemented in the system chain in the near future.

For the second concern, the modulation speed in our design is limited by the response time of the element as well as the nonlinearity of the controlling circuit of the meta-atom. In order to increase the response speed of the element, we can use high-speed PIN diodes in the future with sharper rising and falling edges. Also, we can further improve the controlling circuit to allow faster modulation frequency. We estimate that the modulation frequency can be increased to nearly several MHz if the two issues mentioned above are addressed. For the current system with low modulation speed, we think that it is suitable for the Internet of Thing (IoT) systems that do not need large data transmissions.

Comment:

Q1. As illustrated in Fig. 2, the modulation frequencies of the two partitions are $\Delta f_1=100$ kHz and $\Delta f_2=200$ kHz, however, the duration (defined as $1/\Delta f$) of Partition 2# is clearly larger than

that of Partition 1# in Fig. 2(d)-(f). Is there any problem with the corresponding time-varying reflection coefficients for different frequency components? Please explain.

Response:

Thank you for your good question. We apologize for the mistake of the durations of the time-varying reflection coefficients in Fig. 2. We have revised the durations of two partitions in Fig. 2 in the revised manuscript (also see Fig. R3 below).

Fig. R3. Phasor diagram for the frequency components f_1 (black arrow line) and f_2 (red arrow line) with the complex amplitudes (a) $A_{f_1}=e^{i\pi/4}$, $A_{f_2}=e^{-i\pi/4}$, (b) $A_{f_1}=\frac{1}{2}e^{i\pi/4}$, $A_{f_2}=e^{i\pi/2}$, (c) $A_{f_1}=\frac{1}{2}e^{-i\pi/4}$, $A_{f_2}=\frac{1}{2}e^{i\pi/2}$. (d-f) The corresponding time-varying reflection coefficients of Partitions 1# and 2# for the manipulations of f_1 and f_2 respectively. The incident frequency is 4.25 GHz, and the modulation frequencies of the two regions are $\Delta f_1=100$ kHz and $\Delta f_2=200$ kHz, respectively. The generated harmonic orders are -1st and +1st in Partitions 1# and 2#, and the corresponding frequencies are $f_1=4.2499$ GHz and $f_2=4.2502$ GHz.

Comment:

Q2. The scale values of the reflection phase in Fig. 5(c) are not identical with the description of “Note that the reflection phase range is beyond 2π .” The authors should check the figures carefully.

Response:

Thank you very much for your careful reading. We are sorry that the unit of the reflection phase is degree instead of radian in Fig. 5(c) in the original manuscript. We have updated it as radian in Fig. 5(c) (also see Fig. R4 below) in the revised manuscript.

Fig. R4. Measured amplitude and phase spectra of the metasurface as the external bias voltage changes.

Comment:

Q3. According to the article, a single column of the proposed ASTCM can be modulated to generate a harmonic signal. However, in general, subwavelength elements should be periodically arranged to a certain size, e.g., several wavelengths, to realize the ideal modulation of electromagnetic waves. Is there any theoretical support for using a unitary structure of less than half a wavelength to achieve precise harmonic modulation? In addition, the adjacent columns are loaded with different modulations to generate harmonics of different frequencies. With such a small interval between adjacent columns, will there be frequency mixing and phase distortion between the generated harmonics? Please explain the problems in detail.

Response:

Thank you for your constructive comment.

Firstly, although the **width** of the single column is shorter than half wavelength, its **length** is larger than a wavelength (1.36λ), which behaves like a wire antenna. To demonstrate the fact that one column of the metasurface can be efficiently employed to achieve precise harmonic modulations, we use a horn antenna as the excitation, and the incident waves are converted to QPSK signals at the column surface (see Fig. R5(a)) with the 1st order harmonic as the carrier wave. The grey columns in Fig. R5(a) are covered by absorbing materials, while the brown column is uncovered for signal modulations. The picture under transmission and the experimental scenario can be found in Figs. R5(b) and (c). Since the QPSK symbols modulated by the metasurface contain both amplitude and phase information, if they are recovered correctly at the receiving end, we can prove that the nonlinear amplitude/phase modulations are accurate in the presence of the single column. In Figs. R6(a) and (b), we observe very good constellations and data transmissions by comparing the transmitted picture and the received picture.

Fig. R5. (a) The brown column of the metasurface is used for modulation, while the grey columns are covered by absorbing materials. (b) The picture under transmission. (c) The experimental scenario.

Fig. R6. The constellation and the recovered picture when only one meta-column is enabled.

Secondly, we investigate the possibility of frequency mixing and phase distortion between the dual harmonics generated by two adjacent columns, as suggested by the referee. We employ two columns of the ASTCM to transmit two pictures (see Figs. R7(a-b)) at 4.2505 GHz and 4.2502 GHz respectively. The experimental scenario is shown in Fig. R7(c).

Fig. R7. (a-b) Two pictures under transmission at 4.2505 GHz and 4.2502 GHz respectively. (c) The experimental scenario.

Here we consider two columns with different distances (d), as illustrated in Figs. R8(a) and (b). When the two columns are adjacent ($d = 24$ mm) in experiments, the measured constellation diagrams at 4.2505GHz and 4.2502GHz are plotted in Figs. R9(a) and (b), respectively. For comparison, Figs. R9(c) and (d) show the corresponding measured constellation diagrams with $d = 360$ mm at the frequencies of 4.2505 GHz and 4.2502 GHz, respectively. It is clear that with the decrease of column distance, the Error Vector Magnitudes (EVMs) of the constellation points will also increase as expected. However, even when the two columns are adjacent, the amplitude and phase distortions are still acceptable for data transmissions.

Fig. R8. Two columns for independent data transmitting, which are marked in dark green and orange. The remaining grey columns are disabled. (a) The two columns are adjacent. (b) The distance of the two columns is 360 mm.

Fig. R9. (a-b) Constellations and corresponding recovered pictures at 4.2505 GHz and 4.2502 GHz when two meta-columns are adjacent. (c-d) Constellations and corresponding recovered pictures at 4.2505 GHz and 4.2502 GHz when two meta-columns are not adjacent.

Comment:

Q4. The independence of the far-field patterns generated on each harmonic is crucial for the communication channels. Fig. 3 and Fig. 4 show the far-field patterns of different harmonic modulations. However, only one case is given as a demonstration for each scenario of different harmonic combinations. The authors should provide more cases to demonstrate that the far-

field beams generated by each harmonic are independent of each other when the metasurface is divided into several partitions.

Response:

Thank you for your valuable suggestions. To demonstrate that the far-field beams generated by different harmonics are independent of each other when the metasurface is divided into several partitions, we have added two cases in Fig. 4 (also see Fig. R10 below) of the revised manuscript.

As shown in Fig. R10(a), the metasurface is divided into four partitions with modulation frequencies $f_1 = 100$ kHz, $f_2 = 200$ kHz, $f_3 = 300$ kHz, and $f_4 = 400$ kHz, respectively. At first, the main lobes of the scattering patterns of the four partitions are respectively designed to point at -60° , -30° , 0° , and 50° by independently designing the phase gradients of the four partitions. Since different modulation frequencies are applied to the four partitions, the scattering patterns can be individually controlled.

To demonstrate the fact that the far-field beams generated by the harmonics are independent of each other, in Fig. R10(b), by revising the phase gradient in the f_3 partition, we can change the main-lobe angle from 0° to 17° . Moreover, by changing the phase gradient of the f_1 partition, the main-lobe angle can be changed to -44° , as shown in Fig. R10(c).

In fact, the metasurface can be further divided into eight partitions by using the modulation frequencies $f_1 = 100$ kHz, $f_2 = 200$ kHz, $f_3 = 300$ kHz, $f_4 = 400$ kHz, $f_5 = 500$ kHz, $f_6 = 600$ kHz, $f_7 = 700$ kHz, and $f_8 = 800$ kHz. Initially, the main lobes of the scattering patterns of the eight partitions are pointed at -60° , -50° , -30° , -10° , 0° , 20° , 40° , and 60° , which are illustrated in Fig. R10(d). Similarly, as shown in Fig. R10(e), to independently steer the main lobe of the f_5 partition from 0° to 7° , only the phase gradient of the f_5 partition needs to be reset. As shown in Fig. S10(f), to independently steer the main lobe of the f_1 partition from -60° to -65° , the phase gradient of the f_1 partition needs to be revised.

Fig. R10. Scattering patterns at multiple frequencies generated by the ASTCM. (a-c) The metasurface is divided into four partitions with the modulation frequencies of $f_1 = 100$ kHz, $f_2 = 200$ kHz, $f_3 = 300$ kHz, and $f_4 = 400$ kHz. (d-f) The metasurface is divided into eight partitions with the modulation frequencies of $f_1 = 100$ kHz, $f_2 = 200$ kHz, $f_3 = 300$ kHz, $f_4 = 400$ kHz, $f_5 = 500$ kHz, $f_6 = 600$ kHz, $f_7 = 700$ kHz, and $f_8 = 800$ kHz. The main lobes of the four-partition case and eight-partition case are directed at different angles. (a) $f_1 \sim f_4$: $-60^\circ, -30^\circ, 0^\circ, 50^\circ$ (b) $f_1 \sim f_4$: $-60^\circ, -30^\circ, 17^\circ, 50^\circ$ (c) $f_1 \sim f_4$: $-44^\circ, -30^\circ, 0^\circ, 50^\circ$. (d) $f_1 \sim f_8$: $-60^\circ, -50^\circ, -30^\circ, -10^\circ, 0^\circ, 20^\circ, 40^\circ, 60^\circ$ (e) $f_1 \sim f_8$: $-60^\circ, -50^\circ, -30^\circ, -10^\circ, 7^\circ, 20^\circ, 40^\circ, 60^\circ$ (f) $f_1 \sim f_8$: $-65^\circ, -50^\circ, -30^\circ, -10^\circ, 0^\circ, 20^\circ, 40^\circ, 60^\circ$.

Comment:

Q5. The measured results of the far-field patterns on different harmonics should be given to validate the proposed strategy.

Response:

Thank you for your constructive suggestion. We have measured the harmonic far-field patterns in the microwave anechoic chamber, as suggested by the referee. The metasurface consists of 16 columns with 24 mm spacing between two adjacent columns. Here, the metasurface is equally divided into two partitions by modulating with $f_1 = 100$ kHz and $f_2 = 200$ kHz, respectively. The frequency of incident plane waves is 4.25 GHz.

Fig. R11. Measured and calculated scattering patterns from the two partitions at 4.2501 GHz and 4.2502 GHz respectively. (a) The main lobe of the scattering pattern from the f_1 partition is directed at -44° . (b) The main lobe of the scattering pattern from the f_2 partition is directed at 44° . (c) The main lobe of the scattering pattern from the f_1 partition is directed at -21° . (d) The main lobe of the scattering pattern from the f_2 partition is directed at 21° .

In Fig. R11, we show four cases where different partitioning strategies are considered, in which the dark green and orange regions are modulated by the frequencies f_1 and f_2 respectively, while the grey columns are unmodulated in the experiment. From Figs. R11(a-d), we can see that the measured results agree well with the simulations, and thus validating our theoretical analysis in this paper. The measured results are added in the revised manuscript.

Comment:

Q6. Are the eight channels applied to transmit pictures in experiment established at different deflection angles on each harmonic or at the specular reflection angles of different harmonics? The authors should clarify it in the article. If the channels of eight harmonics are established at different deflection angles, please add the measured results of far-field patterns of the different harmonics to demonstrate the feasibility of QPSK. If the channel of each harmonic is established at the angle of specular reflection, how to transmit eight different data streams at deflected angles? And is it possible to transmit different pictures in eight deflected-angle channels with QPSK modulation?

Response:

Thank you for your valuable comment. Theoretically, since we can control the phase gradient of the internal partitions, the pictures can be transmitted to users at different deflection angles. However, in our experiment, due to the limitation of the metasurface size (16 columns), we cannot achieve beamforming at arbitrary angles. Here, we enabled eight transmission channels since each channel is only controlled by two columns. Hence, we received the information on different channels at specular reflection angles, as pointed out by the reviewer.

To prove the capability that we can transmit different pictures to different deflection angles, we use two channels with the modulation frequencies of $f_1 = 200$ kHz and $f_2 = 500$ kHz. The operating frequency is 4.25 GHz. The two channels can be controlled by the left and right eight columns in the experiments respectively, enabling the independent beam steering by changing

the phase gradients inside each partition. Here we implement two sets of scattering patterns. The main lobe of one scattering pattern points at around 0° , while the main lobe of the other scattering pattern points at around -44° , which is illustrated in Fig. R11(a).

The experimental scenario is illustrated in Fig. R12. Specifically, the ASTCM was illuminated by the transmitting horn which was connected to a microwave signal generator (Keysight E8267D). The incident frequency was set at 4.25 GHz based on the performance of the ASTCM. The ASTCM was modulated with a controlling platform (PXIe-1082, NI Corp.) consisting of a high-speed I/O bus controller, an FPGA module, a digital-analog conversion module, a DC power supply module, and a timing module, which enables us to provide high-quality biasing signals on the embedded varactor diodes within the metasurface. In the receiving end, two receiving antennas (Receiver A and Receiver B) were employed to record the echo waves from the metasurface with a software-defined radio transceiver (NI USRP RIO 2943R). Receiver A was located in the direction normal to the ASTCM, while Receiver B was deviated -44° from the normal direction. The distance between the ASTCM and the receiving antennas is 3 m.

Fig. R12. Experimental scenario of data transmission at two frequencies toward different deflection angles.

Then we investigate three cases as illustrated in Fig. R13(a), Fig. R14(a), and Fig. R15(a), where different main-lobe angles at 4.2502 GHz and 4.2505 GHz are considered. During the whole experiments, the two partitions of the ASTCM are enabled simultaneously.

Fig. R13. (a) The main lobes of the scattering patterns from f_1 and f_2 partitions are both directed at 0° . (b) Measured constellations and recovered pictures from Receiver A at the frequencies of 4.2502 GHz (top) and 4.2505 GHz (bottom). (c) Measured constellations and recovered pictures from Receiver B at the frequencies of 4.2502 GHz (top) and 4.2505 GHz (bottom).

In Case 1 (see Fig. R13(a)), the main lobes of the scattering patterns are both directed to 0° at 4.2502 GHz and 4.2505 GHz. Receivers A and B are located at 0° and -44° , respectively. As shown in Fig. R13(b), the constellations from Receiver A at dual frequencies are distinguishable. In contrast, the quality of the constellations at 4.2502 GHz and 4.2505 GHz greatly deteriorates owing to the deviation angle between Receivers A and B. Only the RF energy from the sidelobe of the scattering waves at the metasurface can be obtained by Receiver B, and thus the corresponding recovered pictures are blurred, as depicted in Fig. R13(c).

In Case 2 (see Fig. R14(a)), the main-lobe angle is 0° at 4.2502 GHz and -44° at 4.2505 GHz. Receivers A and B are located at 0° and -44° , respectively. Fig. R14(b) shows that the constellation from Receiver A at 4.2502 GHz is quite good. However, it is poor at 4.2505 GHz at the presence of the deviation angle between Receiver A and the scattering main lobe at 4.2505 GHz. Similarly, Fig. S14(c) shows the poor constellation from Receiver B at 4.2502 GHz and the good one at 4.2505 GHz.

Fig. R14. (a) The main lobes of the scattering patterns from f_1 and f_2 partitions are directed at 0° and -44° respectively. (b) Measured constellations and recovered pictures from Receiver A at 4.2502 GHz (top) and 4.2505 GHz (bottom). (c) Measured constellations and recovered pictures from Receiver B at 4.2502 GHz (top) and 4.2505 GHz (bottom).

Fig. R15. (a) The main lobe of the scattering pattern from f_1 and f_2 partitions are directed at -44° and 0° directly. (b) Measured constellations and recovered pictures from Receiver A at 4.2502 GHz (top) and 4.2505 GHz (bottom). (c) Measured constellations and recovered pictures from Receiver B at 4.2502 GHz (top) and 4.2505 GHz (bottom).

In Case 3 (Fig. R15 (a)), the main-lobe angle is -44° at 4.2502 GHz and 0° at 4.2505 GHz, just opposite to Case 2. Receivers A and B are located at 0° and -44° , respectively. Fig. R15(b) shows that the constellation from Receiver A at 4.2502 GHz is quite poor, but at 4.2505 GHz it is very good, as expected. On the contrary, Fig. R15(c) shows that the recovered picture and constellation from Receiver B are good at 4.2502 GHz but poor at 4.2505 GHz.

To sum up, the measured results show that the ASTCM has the ability to transmit different information to different deflection angles. These results have been added to the **Supplementary Information**.

To Referee #3:

Comment:

The submitted manuscript presents an asynchronously modulated metasurface for frequency division multiplexing of information. The presented results are interesting and can be of interest across different disciplines. I believe the following (in order of appearance) need to be addressed before this paper is acceptable for publication:

Response:

Thank you very much for the positive comments. We have addressed all your concerns, as detailed in the following responses.

Comment:

Q1. In the current format, the manuscript shares too much similarity with reference 42. One may argue that up to 2.2 (page 12), the material is presented in Ref 42 as well. Please provide a clarification on the novelty and contribution of this work over 42.

Response:

Thank you very much for your insightful comments. In Ref. 42, the ASTCM was initially proposed and developed for generating multiple frequencies under the same excitation. However, how to effectively control the electromagnetic waves and signals at the multiple frequencies has not been investigated in Ref. 42, which remains a great challenge for the application of the asynchronously modulated metasurface. That is the starting point of the current paper. Compared to Ref. 42, the main contributions of this work can be summarized into three parts:

- 1) We propose an efficient way to control the amplitude and phase responses of multiple frequencies generated by the ASTCM simultaneously and accurately.
- 2) The concept of partition multiplexing is firstly proposed to realize the multiple frequency beamforming within a shared aperture.
- 3) We have finished the first experimental demonstration of eight-channel data transmissions with single ASTCM under the excitation of a monochromatic wave. According to the papers published in literatures, the number of independently transmitted channels based on the metasurface were no more than four.

Comment:

Q2. On page 2, the authors refer to "light-matter" interaction, while this work uses RF signals. Please reword.

Response:

Thank you very much for the valuable suggestion. The "light-matter" interaction is replaced by "wave-matter-information" interaction in the revised manuscript, since the digital information contained in the information metasurface is also modulated.

Comment:

Q3. The phrase "the quantification bit number of the reflection phase" on page 3 is not clear. Before that, no mention of quantification but number, reflection phase are presented. Please reword.

Response:

Thank you very much for the good comment.

In Ref. 41, the authors quantified the reflection phase states of the metasurface in digital bits. For instance, a 1-bit coding metasurface is constructed by a sequence of two coding elements "0" and "1", exhibiting 0° and 180° phase responses, respectively. To avoid the unclarity, in

the revised manuscript, we have added the following expressions: “In general, the continuous reflection phase is discretized into finite phase states with the phase step of $2\pi/2^n$, where n is the quantization bit number.”

Comment:

Q4. What is Gamma in equation 4?

Response:

Thank you very much for your kind reminder. Gamma represents the time-varying reflectivity of the metasurface. The definition is added in line 8, paragraph 2, page 5 of the revised paper.

Comment:

Q5. On page 7, the authors state that we would need a large number of PIN diodes to achieve desired phase states. If instead of PIN diodes, authors had used varactor diodes, would that be solved?

Response:

Thank you very much for your good question. Actually, the PIN diodes can be replaced by the varactor diodes in the meta-atom designs to get more phase states, but it will lead to two drawbacks: 1) The bandwidth of the varactor-loaded metasurface is quite limited, which is not favored for wideband applications; 2) The controlling systems of the meta-atoms will be more complex since the biasing voltages of the varactors are usually much higher than that of the PIN diodes. Therefore, we need to use additional driving circuit for each meta-atom to realize large phase range.

Comment:

Q6. The radiation patterns in figure 3 and 4 are presented without any scale. As a result, we cannot deduce the level of sidelobes. Please fix that.

Response:

Thank you very much for your good suggestion. In the revised version, the scales are added to the patterns in Figs. 3 and 4, which are also illustrated in Fig. R16 and Fig. R17 below.

Fig. R16. Measured and calculated scattering patterns respectively generated by two partitions at 4.2501 GHz and 4.2502 GHz. (a) The main lobe of the scattering pattern from the f_1 partition is directed at -44° . (b) The main lobe of the scattering pattern from the f_2 partition is directed at 44° . (c) The main lobe of the scattering pattern from the f_1 partition is directed at -21° . (d) The main lobe of the scattering pattern from the f_2 partition is directed at 21° .

Fig. R17. Scattering patterns of multiple frequencies generated by the ASTCM. (a-c) The metasurface is divided into four partitions with the modulation frequencies of $f_1 = 100$ kHz, $f_2 = 200$ kHz, $f_3 = 300$ kHz, and $f_4 = 400$ kHz. (d-f) The metasurface is divided into eight partitions with the modulation frequencies of $f_1 = 100$ kHz, $f_2 = 200$ kHz, $f_3 = 300$ kHz, $f_4 = 400$ kHz, $f_5 = 500$ kHz, $f_6 = 600$ kHz, $f_7 = 700$ kHz, and $f_8 = 800$ kHz. The main lobes from the four-partition and eight-partition cases are designed toward different angles. (a) $f_1 \sim f_4$: $60^\circ, -30^\circ, 0^\circ, 50^\circ$ (b) $f_1 \sim f_4$: $-60^\circ, -30^\circ, 17^\circ, 50^\circ$ (c) $f_1 \sim f_4$: $-44^\circ, -30^\circ, 0^\circ, 50^\circ$. (d) $f_1 \sim f_8$: $-60^\circ, -50^\circ, -30^\circ, -10^\circ, 0^\circ, 20^\circ, 40^\circ, 60^\circ$ (e) $f_1 \sim f_8$: $-60^\circ, -50^\circ, -30^\circ, -10^\circ, 7^\circ, 20^\circ, 40^\circ, 60^\circ$ (f) $f_1 \sim f_8$: $-65^\circ, -50^\circ, -30^\circ, -10^\circ, 0^\circ, 20^\circ, 40^\circ, 60^\circ$.

Comment:

Q7. Can this device steer the reflected beam toward arbitrary angles or specific angles? The results in fig 4 are presented for increments of 10 degrees.

Response:

Thank you very much for your good question. The device can steer the reflected beam toward arbitrary angles. For example, if the desired beam angle is θ_0 , the reflection phase of the n^{th} meta-column can be given by:

$$\varphi_n = -kd(n - 1) \sin \theta_0, \quad (\text{S2})$$

where k and d represent the wavenumber of the desired harmonic frequency and the period of the meta-atom, respectively.

Comment:

Q8. In figure 5c, the range of reflection angles are only 6 degrees. Is that intended? Please elaborate on how the amplitude and phase been measured.

Response:

Thank you for your good question. We apologize for the mistake. The range of the reflection phase is over 6 rad instead of 6 degrees. We have corrected it in the revised paper, which can also be seen in Fig. R18.

The flow of the reflection amplitude and phase measurement is summarized as follows. Firstly, the metasurface is controlled by external bias voltages output from a DC Power Supply (IT6720). Then, a horn antenna connected to a vector network analyzer (PNA-L Network Analyzer, N5230C) is placed in the far field of the metasurface. As the bias voltage changes, the S_{11} measured by the VNA are recorded. The amplitude and phase of S_{11} are separated, which corresponds to the reflection amplitude and phase of the metasurface respectively.

Fig. R18. Measured amplitude and phase spectra of the metasurface as the external bias voltage changes.

Comment:

Q9. What is the interference between different bands? It seems the patterns generated in figure 4 should result in interference between different bands.

Response:

Thank you for your good comment. To investigate the interference of patterns with different frequencies, we conduct an experiment in the microwave anechoic chamber. The metasurface is equally divided into left and right partitions and modulated with the frequencies of $f_1 = 100$ kHz and $f_2 = 200$ kHz, respectively. The main lobe of the scattering pattern from f_1 partition is designed to be -44° , while the main lobe of the scattering pattern from f_2 partition is designed to be 44° .

Firstly, the left and right partitions are modulated alternatively. The scattering patterns of the 1st order harmonic from dual partitions are shown in Figs. R19(a) and (b) respectively. Then the two partitions are modulated simultaneously. Figs. R19(c) and (d) illustrate the scattering patterns from dual partitions for comparison. It can be seen that the interference does exist,

which results in slight deviation of the original scattering patterns at different bands. But the influence is still limited. Hence, the independent data transmissions at eight channels are not severely affected. These results have been added to the **Supplementary Information**.

Fig. R19. (a-b) The measured scattering patterns from f_1 (dark green) and f_2 (orange) partitions when the left and right partitions are modulated alternatively. (c-d) The measured patterns from f_1 and f_2 partitions when dual partitions are modulated simultaneously.

Comment:

Q10. When we partition the aperture the way the authors suggest, we also reduce the directivity. Does that reduce the SNR? If yes, have you measured it? This is an important observation. Because now we have a device that has lower SNR and increased interference (see previous comment). So these observations and studies are important to compare this device with traditional counterparts.

Response:

Thank you for your valuable comment. We measured the Signal-to-Noise Ratio (SNR) when the entire metasurface, two meta-columns, and single meta-column are employed to transmit one data stream, respectively. Fig. R20 provides the experiment setup, where the ASTCM was illuminated by the transmitting horn connected to a microwave signal generator (Keysight E8267D). The incident frequency was set to 4.25 GHz. The ASTCM was modulated with the controlling platform (PXIe-1082, NI Corp.), which consists of a high-speed I/O bus controller, an FPGA module, a digital-analog conversion module, a DC power supply module, and a timing module, which enables us to provide high-quality biasing signals on the embedded varactor diodes within the metasurface. At the receiving end, the receiving antenna was employed to record the echo waves from the metasurface with a software-defined radio transceiver (NI USRP RIO 2943R).

Fig. R20. Experimental scenario of SNR measurement

The SNR results are 41.69 dB (the entire metasurface), 36.78 dB (two meta-columns), and 32.08 dB (single meta-column), indicating that partitioning the aperture will reduce the SNR as predicted. In practical applications, we can increase the metasurface size and use more columns for one frequency channel to enhance the SNR in wireless communications. These results have been added to the **Supplementary Information**.

REVIEWER COMMENTS

Reviewer #1 (Remarks to the Author):

My concerns have been addressed in the response letter, and I recommend publication of this work in NC.

Reviewer #2 (Remarks to the Author):

The authors have responded and revised the manuscript based on my previous concerns, and the content has been much improved. However, I still not quite satisfy with some of the responses to my comments therefore may need the authors to pay more attentions. Fore my first concern about whether the ASTCM can be work for the receiver or not, the answer seems that they still need horn antenna connected by conventional signal channel receiver, therefore such configuration does not surpass the ordinary RF receiver system with regards to the complexity. It is recommend that the authors make more clear clarification and detailed explanation about how to use the ASTCM for receiving. I believe this would be quite important to this proposed new concept and much of the readers may be interested in knowing this.

Besides, I think the authors may consider the following additional technical issues and make proper responses:

1. Although the authors provide experimental verification of using a unitary structure of less than half a wavelength to achieve precise harmonic modulation in the Response to Comment 3, Referee 2, can the authors add some theoretical analysis? Is there any limitation of the structure size to perform well in precise harmonic modulation?
2. As the authors claimed that the DC signal of the baseband spectra is originated from the unmodulated columns, then why the DC signal in Figure 6 has almost identical amplitude with other frequency components, as the numbers of corresponding columns are obviously different?
3. According to Figure 6c, the normalized amplitude of different frequency component increases as the frequency increases, however, the constellation diagrams at f_1 and f_4 are more dispersed than that in f_2 and f_3 . Please add some discussions.

Reviewer #3 (Remarks to the Author):

The authors have to some extent addressed my concerns. However, they need to answer the following issues before this paper is published:

1) Figure 1, while beautiful, is misleading. The proposed configuration cannot be divided in both dimensions. Please replace it with a more accurate representation of the proposed work.

2) What is the theoretical limit on the number of channels that can be formed? In other words, why stop at 8 (if the hardware was available).

3) I find this sentence a bit confusing: "It is worth noting that the proposed dual harmonic manipulation scheme based on the space-dependent reflectivity functions described in Eq. (7) is especially advantageous to the previous one, which relies on the same reflectivity function within the whole metasurface and leads to the severe entanglement of two harmonics" The authors try to explain this point by the case of equation 8. But that is not very useful. I think they should use the case where the two partitions are used for generation of different frequencies and then show that by adjusting the time delay Δt and M , they can overcome the entanglement of the harmonics. In other words, (8) is not an apple to apple comparison to (7).

4) Related to the previous comment, it is very important the authors elaborate better on the role of Δt and its intelligent selection. To me, that seems to be a major change to reference 42. So elaborating on the selection of M_1 , M_2 , Δt_1 , Δt_2 , and how their selection matters is very useful.

5) The authors state that "The measured patterns of the metasurface with the two partitions operating simultaneously can be found in the..." This gives the impression that none of the results presented in the main part are for simultaneous cases. Is that correct? If yes, please specifically say that in the manuscript.

6) Related to the previous comment, are the results of figures 3 and 4 simultaneous modulation?

7) Is figure 4 simulation or measurement? It needs to be specified in the caption.

8) When the structure in the format of figure 3 and 4, what is the reflection pattern for the monochromatic wave at 4.25 GHz? It also helps understand how much of the power has actually been transferred to the harmonic.

9) The authors state "Next, we exploit them as the carrier frequencies to transmit..." please specify them. It is not clear what "them" refer to.

10) The authors state that the wireless communication is happening at "the corresponding specular reflection angles here,.." Would that be normal direction? please specify in the manuscript.

11) In my view, figures S4 and S5 of the supplemental material should be added the manuscript. They capture the utility and limitations of the proposed work much better (impact of interference, division of frequency and deflection angles, etc). If page limitations is an issue, they can replace figure 3 while figure 3 is moved to the supplemental material.

12) When referring to the supplemental material, please be specific. For example, which figures.

Responses to Referees' Comments

The authors appreciate the constructive comments and suggestions from all referees, which help improve the quality of the manuscript and make the manuscript more complete. We have revised the original manuscript carefully based on these comments, and the revised parts are marked with red fonts. Below are our replies (in blue fonts) to the comments (in black fonts).

To Referee #1:

Comment:

My concerns have been addressed in the response letter, and I recommend publication of this work in NC.

Response:

Thank you very much for your positive comments and kind support.

To Referee #2:

Comment:

The authors have responded and revised the manuscript based on my previous concerns, and the content has been much improved. However, I still not quite satisfy with some of the responses to my comments therefore may need the authors to pay more attentions. Fore my first concern about whether the ASTCM can be work for the receiver or not, the answer seems that they still need horn antenna connected by conventional signal channel receiver, therefore such configuration does not surpass the ordinary RF receiver system with regards to the complexity. It is recommend that the authors make more clear clarification and detailed explanation about how to use the ASTCM for receiving. I believe this would be quite important to this proposed new concept and much of the readers may be interested in knowing this.

Response:

Thank you very much for your good question. As shown in Fig. R1, under the illumination of external RF signals, the diodes of the meta-atoms are biased by periodic voltage waveforms. The impedance of meta-atom becomes a function of the modulation frequency and the voltage $Z(f)$. Thereby the reflection coefficient of the meta-atom can be written as:

Fig. R1. Diagram of the ASTCM-based down-conversion receiver

$$\Gamma(f) = \frac{Z(f) - Z_0}{Z(f) + Z_0}, \quad (R1)$$

where Z_0 is free-space wave impedance. Then the reflected electric field E_r becomes:

$$E_r = E_i \cdot \Gamma(f). \quad (R2)$$

It has been proved that the spectrum of the reflected field can be manipulated by changing the periodic modulation waveform of the meta-atom. That is to say, we can realize up-conversion or down-conversion of the incident field E_i directly on the metasurface without using the traditional RF chain. As illustrated in the inset of Fig. R1, for the metasurface-based receiver, the down-conversion should be employed and the corresponding IF signal will be received by an LC filter near the element. Then the IF signal will be sampled by the ADC under the element and sent to DSP for further

postprocessing. In this way, we can get the received signals from each element. When different partitions are used to record the waves from various RF channels, the signals sensed by each meta-atom can be collected efficiently in the baseband.

In 2020, we tried to realize the down-conversion with the metasurface. An experiment has been carried out and reported in *IEEE Wireless Communications*, vol. 27, no. 2, pp. 180-187. In that paper, the carrier frequency of the RF signal was slightly down shifted, since it still remained a great challenge to generate high-frequency modulation pulses with a period below 1ns. Therefore, we used the traditional RF chains in that paper for receiving purposes.

Now we are still trying to improve the modulation frequency so that the RF signal can be directly down-converted to the IF signal for ADC sampling. We hope that we can develop the metasurface receiver in the near future in our lab.

Besides, I think the authors may consider the following additional technical issues and make proper responses:

Comment:

Q1. Although the authors provide experimental verification of using a unitary structure of less than half a wavelength to achieve precise harmonic modulation in the Response to Comment 3, Referee 2, can the authors add some theoretical analysis? Is there any limitation of the structure size to perform well in precise harmonic modulation?

Response:

Thank you very much for your insightful question. The length of a meta-column as the smallest structure for harmonic modulation is 1.36λ in our paper. Therefore, we believe that it should not be considered a structure of less than half wavelength.

In theory, we can regard each meta-atom as an electrically small antenna that can generate dynamic amplitude and phase responses required by the harmonic modulation

under the illumination of RF signals, and then radiate the energy back into free space. Moreover, in practical applications, a number of meta-atoms can be employed to ensure the desired scattering pattern besides the harmonic modulation.

To demonstrate the dependence of the modulation ability on structure size, we provide the simulated phase responses when different numbers of meta-atoms are excited by plane waves. Open space boundary conditions are used in the simulation. Fig. R2 shows the reflected phases with the change of meta-atom numbers when the varactor diode is biased at 0 and 6V, respectively. From the phase curves of the current element in Fig. R2, we observe that the phases are relatively stable when the element number is greater than 4.

Fig. R2. Phase response as a function of the meta-atom number when the embedded varactor diode is biased at 0V and 6V respectively.

Comment:

Q2. As the authors claimed that the DC signal of the baseband spectra is originated from the unmodulated columns, then why the DC signal in Figure 6 has almost identical amplitude with other frequency components, as the numbers of corresponding columns are obviously different?

Response:

Thank you very much for your good question. We carried out the experiment again and found that the phenomenon is connected to the relative positions of the sample, the transmitting antenna, and receiving antenna. Fig. R3(a) shows the overall configuration of our experiment, in which we considered three cases when the receiving antenna slightly moves to the left and right, as illustrated in Figs. R3(b-d). The corresponding measured baseband spectra are presented in Figs. R4(a-c) when the receiving antenna is placed at different positions. We clearly observe significant changes in the relative harmonic magnitudes, which may be ascribed to the spatial multipath and shading effect when the two antennas and the sample are closely deployed. Therefore, although the areas of the high-order harmonics are limited, their magnitudes are somehow quite large due to the multiple reflections on the surfaces of the surrounding scatterers.

Fig. R3. (a) Configuration of the experiment. (b-d) Positions of the receiving antenna relative to the ASTCM. (b) The receiving antenna is just behind the transmitting antenna. (c) The receiving antenna is on the left of the transmitting antenna. (d) The receiving antenna is on the right of the transmitting antenna.

Fig.R4. The received baseband spectra when the receiving antenna is placed at different positions. (a) The receiving antenna is just behind the transmitting antenna; (b) The receiving antenna is on the left of the transmitting antenna; (c) The receiving antenna is on the right of the transmitting antenna.

Comment:

Q3. According to Figure 6c, the normalized amplitude of different frequency component increases as the frequency increases, however, the constellation diagrams at f_1 and f_4 are more dispersed than that in f_2 and f_3 . Please add some discussions.

Response:

Thank you very much for your insightful comment. As stated in the reply of Q2, the relative positions of the sample, transmitting antenna, and receiving antenna are quite critical for the received harmonic energy. In the experiment of Fig. 6c, the relative positions are not identical to those in Fig. 8(a-d). We have measured the constellation diagrams (see Fig. R5) with the same configuration to Fig. 6c. It can be seen that the constellation diagram at f_1 is slightly more dispersed than that in f_2 , f_3 , and f_4 in this case, because the amplitude differences for the four generated frequencies are relatively small.

Fig. R5. The measured constellation diagrams at f_1 , f_2 , f_3 and f_4 with the same configuration as that in Fig. 6c.

To Referee #3:

Comment:

The authors have to some extent addressed my concerns. However, they need to answer the following issues before this paper is published:

Response:

Thank you very much for your good comments.

Comment:

Q1. Figure 1, while beautiful, is misleading. The proposed configuration cannot be divided in both dimensions. Please replace it with a more accurate representation of the proposed work.

Response:

Thank you very much for your good suggestion. Fig. 1 has been revised, as shown in Fig. R6 below.

Fig. R6. Schematic of wave manipulations at multiple frequencies based on the asynchronous space-time-coding digital metasurface, in which the partitions with different colors are responsible for the wave manipulations at different carrier frequencies.

Comment:

Q2. What is the theoretical limit on the number of channels that can be formed? In other words, why stop at 8 (if the hardware was available).

Response:

Thank you very much for your good question. Theoretically, each element can serve as an independent channel for signal modulation. In the experiment, the channel number is limited due to the available DAC numbers and other driven modules in our lab.

Comment:

Q3. I find this sentence a bit confusing: "It is worth noting that the proposed dual harmonic manipulation scheme based on the space-dependent reflectivity functions described in Eq. (7) is especially advantageous to the previous one, which relies on the same reflectivity function within the whole metasurface and leads to the severe entanglement of two harmonics" The authors try to explain this point by the case of equation 8. But that is not very useful. I think they should use the case where the two partitions are used for generation of different frequencies and then show that by adjusting the time delay Δt and M , they can overcome the entanglement of the harmonics. In other words, (8) is not an apple to apple comparison to (7).

Response:

Thank you very much for your good question. We are sorry for the confusion of the sentence. In Eq. (8), we give a traditional way for dual-frequency manipulations with the whole metasurface. Such a strategy is hard to extend for multiple frequency manipulations since we need to find a very complex reflection phase function Γ_0 . However, in Eq. (7), we can use different partitions with independent modulation waveforms to circumvent this problem. As a result, the number of independent frequency channels is greatly improved, which is equal to the partition number within the metasurface. In the revised manuscript, we have deleted this confusing sentence and rewritten the discussion as follows:

“

$$\begin{cases} A_m e^{j(\varphi_m - 2\pi m \Delta f_1 \Delta t_1)} = \frac{1}{T_1} \int_0^{T_1} \Gamma_1(t - \Delta t_1) e^{-j2\pi m \Delta f_1 t} dt \\ A_n e^{j(\varphi_n - 2\pi n \Delta f_2 \Delta t_1)} = \frac{1}{T_2} \int_0^{T_2} \Gamma_2(t - \Delta t_2) e^{-j2\pi n \Delta f_2 t} dt \end{cases} \quad (7)$$

From Eq. (7), it can be seen that the time delays only affect the harmonic phases rather than amplitudes, which enables the separation of the two responses effectively. Thus we are able to synthesize the desired two harmonics at f_1 and f_2 by selecting the combinations of $\Gamma_1(t)$ and $\Gamma_2(t)$ with proper combinations of the duty ratios M_1 , M_2 , and the time delays Δt_1 , Δt_2 .

By letting $\Gamma_1(t) = \Gamma_2(t) = \Gamma_0(t)$, and $T_1 = T_2 = T$ in Eq. (6), the harmonic amplitudes and phases can be expressed as follows:

$$\begin{cases} A_m e^{j\varphi_m} = \frac{1}{T} \int_0^T \Gamma_0(t) e^{-j2\pi m \Delta f t} dt \\ A_n e^{j\varphi_n} = \frac{1}{T} \int_0^T \Gamma_0(t) e^{-j2\pi n \Delta f t} dt \end{cases} \quad (8)$$

In Eq. (8), we give a traditional way for dual frequency manipulations with the whole metasurface. Once the reflectivity function $\Gamma_0(t)$ is chosen for the m^{th} -order harmonic manipulation, the amplitude and phase of the n^{th} order harmonic are also determined as well. In order to independently control the two harmonics with arbitrary A_m , A_n , φ_m and φ_n , it poses great challenges to find proper reflectivity function $\Gamma_0(t)$. In fact, it is hard to find or realize the time-varying reflectivity functions according to the desired amplitude and phase combinations. To circumvent this problem, the intertwined coding sub-sequences were employed to generate and manipulate multiple harmonics [41], but it suffers from three additional major problems: complicated coding algorithms, slow reconfiguration rate, and limited harmonic numbers. In general, the independently controlled harmonic number is equal to the quantized bit width of the reflection phase of the metasurface. This implies that a large number of PIN diodes are demanded in the meta-atoms to achieve more phase states to promote the number of harmonics, resulting in significant increases of system cost and power consumption. Such a strategy is hard to extend for the purpose of multiple frequency manipulations since we need to find a very complex reflection phase function Γ_0 . However, in Eq. (7), we can use different partitions with independent modulation waveforms to circumvent this problem. As a result, the number of independent frequency channels is greatly improved, which is equal to the partition number within the metasurface”.

Comment:

Q4. Related to the previous comment, it is very important the authors elaborate better on the role of Δt and its intelligent selection. To me, that seems to be a major

change to reference 42. So elaborating on the selection of M_1 , M_2 , Δt_1 , Δt_2 , and how their selection matters is very useful.

Response:

Thank you very much for your good question. The selection of M_1 , M_2 , Δt_1 , and Δt_2 for particular amplitudes and phases of the harmonics is as follows:

As derived in the context, the amplitudes and phases of the k^{th} -order harmonic in the partitions 1# and 2# can be written as A_k^1 , φ_k^1 , A_k^2 , φ_k^2 , where the subscript and superscript stand for the harmonic and partition order, respectively. For simplicity, we consider the 1st-order generation in partition 1#:

$$\begin{cases} A_1^1 = 2 \cdot M_1 \cdot \left\lfloor \frac{\sin \pi M_1}{\pi M_1} \right\rfloor \\ \varphi_1^1 = -\frac{\pi}{2} [1 - (-1)^{\lfloor M_1 \rfloor}] \end{cases} \quad (\text{R3})$$

where $\lfloor \cdot \rfloor$ indicates the operation of rounding down. If we want to get the desired amplitude and phase (A_1^1 and φ_1^1) for the 1st-order harmonic, we need to solve this nonlinear equation with the numerical method.

Fig. R7. The relationship between M_1 and the 1st-order harmonic amplitude in partition 1#.

Firstly, from Fig. R7, we can give the relationship between A_1^1 and M_1 . From this figure, we can easily get a proper candidate of M_1 for specific amplitude A_1^1 .

Secondly, we can get the phase φ_1^1 from Eq. (R2) when M_1 is chosen. If there is a deviation $\Delta\varphi$ between the obtained and desired phases, we can use the time delay Δt_1 for compensation:

$$\Delta t_1 = \frac{-\Delta\varphi}{2\pi f_1}, \quad (\text{R4})$$

in which f_1 represents the modulation frequency of partition 1#.

By following the above two steps, we can get the values of M_1 and Δt_1 to obtain the desired harmonic amplitude and phase in partition 1#. The same approach can be used to determine M_2 and Δt_2 for arbitrary A_k^2 , φ_k^2 in partition 2#.

Comment:

Q5. The authors state that "The measured patterns of the metasurface with the two partitions operating simultaneously can be found in the..." This gives the impression that none of the results presented in the main part are for simultaneous cases. Is that correct? If yes, please specifically say that in the manuscript.

Response:

Thank you very much for your kind reminder. For better demonstration, we have displayed both simultaneous and non-simultaneous cases in the main part of the revised manuscript, as seen in Fig. R8. Please also see Fig. 3 in the revised manuscript.

Fig. R8. (a-b) The measured scattering patterns from f_1 (dark green) and f_2 (orange) partitions when the left and right partitions are modulated alternatively. (c-d) The measured patterns from f_1 and f_2 partitions when dual partitions are modulated simultaneously.

Comment:

Q6. Related to the previous comment, are the results of figures 3 and 4 simultaneous modulation?

Response:

Thank you very much for your good question. The results of Fig. 3 in the original version are not simultaneously modulated. For better illustration, in the revised paper, Figs. 3(a) and (b) give the measured results when both partitions are not simultaneously modulated, while Figs. 3(c) and (d) discuss the simultaneously modulated case. In the meanwhile, Fig. 4 shows the ideal simulation results when all partitions are modulated simultaneously.

Comment:

Q7. Is figure 4 simulation or measurement? It needs to be specified in the caption.

Response:

Thank you very much for your kinder reminder. Fig. 4 illustrates the simulated results and we have emphasized this in the caption.

Fig. R9. Simulated multi-frequency scattering patterns of the partition ASTCM. (a-c) The metasurface is divided into four partitions with the modulation frequencies $f_1 = 100$ kHz, $f_2 = 200$ kHz, $f_3 = 300$ kHz, and $f_4 = 400$ kHz. (d-f) The metasurface is divided into eight partitions with the modulation frequencies $f_1 = 100$ kHz, $f_2 = 200$ kHz, $f_3 = 300$ kHz, $f_4 = 400$ kHz, $f_5 = 500$ kHz, $f_6 = 600$ kHz, $f_7 = 700$ kHz, and $f_8 = 800$ kHz. The main lobes of the four-partition case and eight-partition case are directed at different angles. (a) $f_1 \sim f_4$: $-60^\circ, -30^\circ, 0^\circ, 50^\circ$; (b) $f_1 \sim f_4$: $-60^\circ, -30^\circ, 17^\circ, 50^\circ$; (c) $f_1 \sim f_4$: $-44^\circ, -30^\circ, 0^\circ, 50^\circ$. (d) $f_1 \sim f_8$: $-60^\circ, -50^\circ, -30^\circ, -10^\circ, 0^\circ, 20^\circ, 40^\circ, 60^\circ$; (e) $f_1 \sim f_8$: $-60^\circ, -50^\circ, -30^\circ, -10^\circ, 7^\circ, 20^\circ, 40^\circ, 60^\circ$; (f) $f_1 \sim f_8$: $-65^\circ, -50^\circ, -30^\circ, -10^\circ, 0^\circ, 20^\circ, 40^\circ, 60^\circ$.

Comment:

Q8. When the structure in the format of figure 3 and 4, what is the reflection pattern for the monochromatic wave at 4.25 GHz? It also helps understand how much of the power has actually been transferred to the harmonic.

Response:

Thank you very much for your good question. Fig. 4 illustrates the ideal simulation results, so there is no reflection pattern at 4.25GHz. To help understand how much of the power has been transferred to the harmonic as suggested, we have presented the measured fundamental harmonic pattern at 4.25 GHz (in yellow) besides the two high order harmonic patterns at 4.2501 GHz and 4.2502 GHz, as shown in Fig. R10.

Fig. R10. Measured scattering patterns at the frequencies of 4.25 GHz, 4.2501 GHz, and 4.2502 GHz when the dual partitions with different modulation frequencies of $f_1 = 100$ kHz and $f_2 = 200$ kHz operate simultaneously.

Comment:

Q9. The authors state "Next, we exploit them as the carrier frequencies to transmit..." please specify them. It is not clear what "them" refer to.

Response:

Thank you very much for your good question. "them" refers to the eight frequencies generated by ASTCM. We have revised the sentence for better clarification.

Comment:

Q10. The authors state that the wireless communication is happening at "the corresponding specular reflection angles here,..." Would that be normal direction? please specify in the manuscript.

Response:

Thank you very much for your good question. The corresponding specular reflection angles are in the normal direction. We have added the relevant explanations in the revised manuscript.

Comment:

Q11. In my view, figures S4 and S5 of the supplemental material should be added the manuscript. They capture the utility and limitations of the proposed work much better (impact of interference, division of frequency and deflection angles, etc). If page limitations is an issue, they can replace figure 3 while figure 3 is moved to the supplemental material.

Response:

Thank you very much for your good suggestions. We have added Fig. S4 and Fig. S5 to the manuscript and moved Fig. 3 in the previous version to the supplemental material.

Comment:

Q12. When referring to the supplemental material, please be specific. For example, which figures.

Response:

Thank you very much for your good suggestion. We have specified all figure numbers when referring to the supplementary material in the revised manuscript.

REVIEWERS' COMMENTS

Reviewer #2 (Remarks to the Author):

The authors have carefully responded to my comments. I just have one more request that the responses to my comments should be reflected in the revision of the main manuscript.

Reviewer #3 (Remarks to the Author):

The authors have addressed the concerns. I think the manuscript can be published once the following revisions are made:

The discussion about selection of M and Δt (answer to Q4 of Ref 3) should be added to supplemental material.

The information that Fig. 4 shows the ideal simulation results when all partitions are modulated simultaneously.

Responses to Referees' Comments

The authors appreciate the constructive comments and suggestions from all referees, which help improve the quality of the manuscript and make the manuscript more complete. We have revised the original manuscript carefully based on these comments, and the revised parts are marked with **red fonts**. Below are our replies (**in blue fonts**) to the comments (in black fonts).

To Referee #2:

Comment:

The authors have carefully responded to my comments. I just have one more request that the responses to my comments should be reflected in the revision of the main manuscript.

Response:

Thank you very much for your valuable suggestion. The replies to your valuable questions have been reflected in the revised paper as suggested.

Specifically, the discussions on the metasurface-based receiver mentioned by the reviewer have been added in Note 5 of Supplementary Information due to the limit of the figure number; the theoretical analysis of precise harmonic modulation with a unitary structure of less than half wavelength is added in Line 3, Paragraph 1, Page 11 of the revised paper; the origin of the almost identical amplitude among the DC signal and other frequency components in Fig. 6 is discussed in Line 2, Paragraph 1, Page 15 of the revised paper; and the more dispersed constellation diagrams at f_1 and f_4 than those in f_2 and f_3 are analyzed in Line 4, Paragraph 2, Page 17 of the revised paper.

To Referee #3:

Comment:

The authors have addressed the concerns. I think the manuscript can be published once the following revisions are made:

The discussion about selection of M and Δt (answer to Q4 of Ref 3) should be added to supplemental material.

The information that Fig. 4 shows the ideal simulation results when all partitions are modulated simultaneously.

Response:

Thank you very much for your valuable comment and suggestion.

The discussion on selections of M and Δt has been added in Note 1 of Supplementary Information.

We have stressed that Fig. 4 gives the ideal simulation results when all partitions are modulated simultaneously in Line 9, Paragraph 1, Page 11 of the revised paper.